# Theoretical Study of Sphingomyelinases from *Entamoeba histolytica* and *Trichomonas vaginalis* Sheds Light on the Evolution of Enzymes Needed for Survival and Colonization

**DOI:** 10.3390/pathogens14010032

**Published:** 2025-01-05

**Authors:** Fátima Berenice Ramírez-Montiel, Sairy Yarely Andrade-Guillen, Ana Laura Medina-Nieto, Ángeles Rangel-Serrano, José A. Martínez-Álvarez, Javier de la Mora, Naurú Idalia Vargas-Maya, Claudia Leticia Mendoza-Macías, Felipe Padilla-Vaca, Bernardo Franco

**Affiliations:** 1Departamento de Farmacia, División de Ciencias Naturales y Exactas, Universidad de Guanajuato, Noria Alta s/n, Guanajuato 36050, Mexico; fb.ramirez@ugto.mx; 2Departamento de Biología, División de Ciencias Naturales y Exactas, Universidad de Guanajuato, Noria Alta s/n, Guanajuato 36050, Mexico; sy.andradeguillen@ugto.mx (S.Y.A.-G.); martinezjose@ugto.mx (J.A.M.-Á.); cl.mendoza@ugto.mx (C.L.M.-M.); 3Genética Molecular, Instituto de Fisiología Celular, Universidad Nacional Autónoma de México, Mexico City 04510, Mexico

**Keywords:** sphingomyelinases, *Entamoeba histolytica*, *Trichomonas vaginalis*, evolution, structural features, AlphaFold3, membrane repair, hemolysis

## Abstract

The path to survival for pathogenic organisms is not straightforward. Pathogens require a set of enzymes for tissue damage generation and to obtain nourishment, as well as a toolbox full of alternatives to bypass host defense mechanisms. Our group has shown that the parasitic protist *Entamoeba histolytica* encodes for 14 sphingomyelinases (SMases); one of them (acid sphingomyelinase 6, aSMase6) is involved in repairing membrane damage and exhibits hemolytic activity. The enzymatic characterization of aSMase6 has been shown to be activated by magnesium ions but not by zinc, as shown for the human aSMase, and is strongly inhibited by cobalt. However, no structural data are available for the aSMase6 enzyme. In this work, bioinformatic analyses showed that the protist aSMases are diverse enzymes, are evolutionarily related to hemolysins derived from bacteria, and showed a similar overall structure as parasitic, free-living protists and mammalian enzymes. AlphaFold3 models predicted the occupancy of cobalt ions in the active site of the aSMase6 enzyme. Cavity blind docking showed that the substrate is pushed outward of the active site when cobalt is bound instead of magnesium ions. Additionally, the structural models of the aSMase6 of *E. histolytica* showed a loop that is absent from the rest of the aSMases, suggesting that it may be involved in hemolytic activity, as demonstrated experimentally using the recombinant proteins of aSMase4 and aSMase6. *Trichomonas vaginalis* enzymes show a putative transmembrane domain and seem functionally different from *E. histolytica*. This work provides insight into the future biochemical analyses that can show mechanistic features of parasitic protists sphingomyelinases, ultimately rendering these enzymes potential therapeutic targets.

## 1. Introduction

Why would a free-living organism give up its independence and create complicated interactions with another organism that imply establishing relationships requiring specific survival strategies? In the case of symbiosis, a source of evolutionary innovation [1,2], such as bacterial symbionts’ influence on the rise of the allocating organ, like *Vibrio fischeri* with the embryo of the squid *Euprymna scolopes* [3], is a clear effect on the development of both the surface epithelium and the epithelial tissue involved in forming the organ where bacteria reside and, thus, a clear link of the influence of bacterial symbionts on the evolution of its host. Nguyen and van Baalen [2] proposed a model that suggests that organisms encountering a host come at a price for any advantageous trait (such as reproduction). The benefit must be constant, and this constraint will maintain the host and symbiont interaction [2]. Under this model, parasitic organisms have reached the maximum benefit in terms of host-seeking strategies (which is associated with a limited source of free hosts according to the model proposed by Nguyen and van Baalen [2]), thus compromising reproduction in the host environment. However, this comes with a price: pathogenic organisms require a toolbox for host immune evasion and survival in the physiological conditions inside the host. 

The ability of a parasite to establish a successful infection depends on a series of evolutionary adaptations, including the development of diverse strategies to evade and survive the host’s innate cellular and humoral effector mechanisms [4].

The pathogenic protists *Entamoeba histolytica* and *Trichomonas vaginalis* are two of the predominant pathogenic protists and a major threat to human health [5]. In the case of *E. histolytica*, estimates suggest that approximately 10% of the global population harbors this parasite, causing 50 million cases of disease and up to 100,000 deaths each year [6,7,8]. *T. vaginalis* is the most prevalent curable non-viral sexually transmitted pathogen, with approximately 156 million cases annually [9,10].

In the case of *E. histolytica*, several virulence factors and determinants have been identified. Also, many remain to be fully elucidated in how they are involved in pathogenesis, such as motility, adherence, phagocytosis, stress response, metabolic regulation, and signaling [11,12]. Also, other factors, such as the interaction with the intestinal microbiota, modulate the virulence of this organism [13], and perhaps lectins are integral elements of the invasion tools used by *E. histolytica* and other pathogenic amoeba [14]. 

However, the knowledge of the genes involved during the invasion of the human colon varies between strains [15], showing diverse up- and down-regulated genes involved in protein translation, DNA–RNA regulation, cell signaling, proteolysis, lipid metabolism, and cytoskeleton genes. Thibeaux and colleagues [15] showed that, during mucus contact, the main proteins needed for attachment are lectin light subunit 2, fibrinogen binding protein, CP-A7, fatty acid ligase, α-amylase, and UDP-glucose isomerase. Also, epigenetic regulation is involved in regulating small GTPases that are involved in diverse cellular trafficking activities and are associated with the loss of virulence [16,17].

*T. vaginalis* virulence traits have been less explored than other pathogenic protists. Menezes and colleagues [18] point out that, in this organism, several uncharacterized secreted virulence factors are relevant in the pathogenesis of the infection, resulting in limited research focused on elucidating virulence factors in this worldwide pathogen. Additionally, Mercer and Johnson [19] summarize the relevant factors associated with colonizing *T. vaginalis* and several surface proteins and enzymes. The exact mechanism of action inside the host is unknown, but the virulence phenotype is associated with symbionts contributing to pathogenesis [19]. 

In the case of *Trichomonas tenax*, an oral pathogen associated with periodontitis, novel candidate virulence factors, such as carbohydrate hydrolytic enzymes, have been identified using comparative genomics against the *T. vaginalis* genome, which in turn suggests that carbohydrate-active enzymes or CAZymes may be relevant in the host–pathogen associations in *T. vaginalis* [20] and may help bypass microbiota bacterial organisms along with the host cells.

Our group is interested in uncovering novel virulence traits in *E. histolytica* associated with host invasion and survival. The first novel trait identified by our group was reported in 2010 [21]. The striking feature found is that two enzymes (termed EhnSM1 and EhnSM3, of a total of three coding genes by name that were found in the *E. histolytica* genome) were identified in the cytosol, and the activity was stimulated by Mg^2+^ but inhibited by Zn^2+^ ions. Later, EhnSM3 was found to be involved in cell contact-dependent increased hemolytic and cytotoxic activity in cells overexpressing EhnSM3 [22], suggesting a role as a β-toxin, as reported for *Bacillus cereus*. 

Sphingomyelinases (aSMases) are enzymes involved in the membrane repair mechanism via the hydrolysis of sphingomyelin and the release of ceramide via the rapid Ca^2+^-mediated endocytosis of the membrane lesion [23,24,25], where ceramide is the second messenger associated with cellular stress. Recently, a novel mechanism of cell membrane repair involving secreted acid sphingomyelinase (aSMase) has been described in the parasitic protozoan *E. histolytica* [26]. Ramírez-Montiel and colleagues’ work showed that *E. histolytica* repairs its plasma membrane by fusing endosomes and lysosomes, which leads to the secretion of one of the six enzymes this organism encodes [26]. Also, the overexpression of aEhSM6, activated by Mg^2+^, and Co^2+^ inhibiting the enzyme increases the survival of cells exposed to membrane-damaging peptides and proteins, and the six aSMase enzymes increase their expression upon exposure to membrane-damaging agents, as well as oxidative stress. This work sets the foundation that, in pathogenic organisms, SMases are needed for cell membrane repair, and they bypass the damage elicited via immune effectors such as antimicrobial peptides, reactive oxygen, and nitrogen species, as well as the lytic activity of the complement [26]. 

This work, using well-established bioinformatic tools, shows that the enzymes from parasitic protists are, in comparison with enzymes from other organisms, not well conserved at the sequence level but show close evolutionary relationships, even with ancestral forms of these enzymes, such as a bacterial glycerophosphodiester phosphodiesterase. With the use of the novel AlphaFold3 web server, the SMase proteins were modeled in the presence of post-translational modifications (glycosylation and phosphorylation) including ion ligands, showing structural variations, depending on the metal ion bound, and when cavity detection-guided blind docking using sphingomyelin as a substrate showed that the binding was strongly modified, depending on the metal ion modeled in the protein. Overall, this work proposes that the SMases, acid and neutral, are novel virulence-associated traits in two relevant pathogenic protists. 

## 2. Materials and Methods

### 2.1. Sequence Retrieval 

All sequences were retrieved from the UniProt database [27], release version 2024_04, 6 September 2024. The SMases sequences from *E. histolytica* and *T. vaginalis* were retrieved from UniProt and further analyzed using different bioinformatic tools. 

### 2.2. Sequence Analysis

Sequence comparison was carried out with the Clustal Omega Job Dispatcher analysis tool [28] and visualized using the Alignment Viewer, release 1.0 (https://alignmentviewer.org/, accessed on 21 September 2024), and the UMAP sequence analysis plug-in [29]. Weblogo was created with WebLogo3 with the default settings [30].

Phylogenetic analysis was conducted with MEGA, version 11.0.13 [31]. The phylogenetic analysis parameters are indicated in each figure legend, and in each case, a bootstrap of 1000 iterations was used. The groups identified were manually colored. For pairwise distances, the estimation was used to compare pairs of taxa without variance estimates, and the Poisson model was used for substitution types using amino acid residues and uniform rates; pairwise deletion was used for missing data or gaps. The sequence pairwise distances were also calculated with MEGA, version 11.0.13. 

### 2.3. Protein Structure Models

Protein structure prediction was conducted using the AlphaFold3 server (https://golgi.sandbox.google.com/, accessed from 14 August 2024 to 22 October 2024) [32] with the default options. In this work, the models used were automatically ranked as the best via AlphaFold3 since this model shows the highest pTM score above 0.5. This means the overall predicted fold for the complex might be similar to the true structure. In Appendix A, we provide the pTM scores of all the models used in this work, following the same order as in Table 1.

Protein models were generated by adding a metal ion (see text) or post-translational modifications like glycosylation moieties or phosphorylation sites and the predicted structure by binding palmitic to simulate the binding site for a hydrophobic ligand. For the glycosylation prediction, the most abundant Asn-linked modifications are branched glycan chains [33]; the code for adding the glycan chains was as follows: NAG(NAG(MAN(MAN(MAN)(MAN(MAN)(MAN))))) in the predicted glycosylated sites using Protter [34]. Protter was also used to predict the signal peptide and putative membrane-spanning regions, which were confirmed by modeling the protein in the presence of 50 oleic acids in the AlphaFold3 server. Catalytic residues were predicted with ScanProsite [35].

Complementary reference structures were retrieved from the PDB or AlphaFold2 database using the link from the UniProt database. The figures indicate these models with the corresponding database accession numbers. 

The structural alignment of protein models was conducted with US-align [36] using the default options. Structural alignment assessment was determined by the root-mean-square deviation (RMSD) and the average TM score for the shorter protein compared to in the alignment. Visualization of protein models for obtaining individual images arranged as in structural alignments was conducted with UCSF Chimera, v. 1.18, for Windows (64-bit version) [37]. A rainbow color scheme and cartoon representation were used to facilitate the localization of the N- (blue) and C-terminal (red) ends. 

### 2.4. Docking Simulation

Protein models were used to estimate the docking of the substrate. The major challenge is that sphingomyelin contains several rotamers. To bypass such a challenge, a C9H21N2O6P sphingomyelin, CID number 70678679, using the 3D conformer was used. Cavity detection-guided blind docking (CB-Dock2 server) was used with the default settings [38]. The substrate used was downloaded from PubChem as an ‘sdf’ file. Proteins were prepared for docking using the UCSF Chimera Dock Prep tool as described previously [37,39]. The phosphorylated models were not used since they presented stereochemical clashes. Docking results were visualized using PyMOL 4.6.0. 

### 2.5. Sequence Motif Analysis

To assess conserved and potentially unique sequences, amino acid sequences of SMases and bacterial hemolysins were analyzed with GLAM2 using the default options [40]. The ten best-scoring motifs predicted were mapped and identified in the protein sequences using MAST with the default options [41]. 

### 2.6. Recombinant Protein Production and Hemolytic Activity Assay

The two recombinant aSMase (aSMase4 and aSMase6) proteins were expressed in *E. coli* using the following strategy. The two proteins were amplified from cDNA from the *E. histolytica* HM1:IMSS strain, excluding the signal peptides from each open reading frame (ORF). For aSMase4, the protein was amplified from 40 to 1221 bp, and aSMase6 was amplified from 43 to 1265 bp of the coding gene. The oligonucleotides used for aSMase4 were Xo-ASM4-D CGCTCGAGCAATCAATAGAAATTAT, including an *Xho*I restriction site, and HASM4-R CCCAAGCTTTAACATCAGTTTAACAC, including a *Hin*dIII restriction site. For aSMase6, the oligonucleotides used were BaASM6-D CGCGGATCCTTTGAAATGTGGCAACT with a *Bam*HI restriction site and HASM6-R CCCAAGCTTTTCCAAACTACACA with a *Hind*III restriction site. The PCR products were digested with the corresponding restriction enzymes and cloned into the pRSET-A (Invitrogen, Waltham, MA, USA) plasmid. Recombinant proteins were purified as described previously [26], and activity was verified as described previously [26]. Hemolytic activity was measured as described previously [22]. In brief, human red blood cells (RBCs) were obtained from healthy donors and stored in Alsever, washed with 10 mM of PIPES-Tris buffer, and adjusted to 1.25 × 10^8^ cells/mL; RBC and 3.3 µg of recombinant proteins were mixed in a 96-well plate as triplicates. Commercially available *B. cereus* hemolysin (Sigma-Aldrich, St. Louis, MO, USA) was used as a positive control. Reactions were carried out in 10 mM of PIPES-Tris buffer at pH 5.0 and 7.0 for 90 min, and the result was recorded with a digital camera. 

## 3. Results

Our group has recently reported that one of the acid SMases (aSMase6, UniProt C4LVV3) from the parasite *E. histolytica*, is involved in the repair mechanism of the plasma membrane through various environmental insults [26]. Also, as shown for the human enzyme [42], in *E. histolytica*, it is processed in the C-terminal end, which is consistent with the presence of several cysteine residues [26]. Thus, the aim of this work, by taking advantage of AlphaFold3, was to explore the structural features of the SMases in parasitic protists such as *E. histolytica* and *T. vaginalis* and explore the relationship between the characteristics of aSMase6 and *E. histolytica* that are known so far. We also sought to determine whether this is consistent in another parasite with a different environment, such as *T. cruzi*, that is not associated with mucosal immunity. 

### 3.1. Sequence Homology in Parasitic Organisms’ SMases

The *E. histolytica* has 14 coding sequences for putative SMase enzymes. Previous work has shown that neutral and acidic enzymes are expressed, and enzymatic activity has been demonstrated in *E. histolytica* trophozoites [21,22,26]. However, the sequences seem to have a low identity between them. Appendix A compares the amino acid sequence of SMase enzymes from *E. histolytica* and *T. vaginalis* annotated in UniProt or GenBank. In this figure, the intensity of the color indicates enzymes with homology, while the lighter colors indicate a low identity between them. Also, as shown in Appendix A, Panel B, through pairwise alignments of all the enzymes from *E. histolytica* and *T. vaginalis*, there are no conserved regions between them. In Panel B, the upper panel shown in the MSA color scheme indicates less conservation (the two logos shown indicate the only two regions with relatively conserved residues); however, the lower panel in the hydrophobicity color scheme suggests that the enzymes preserve physicochemical features, which could be associated with conserved folding, too. Therefore, the enzymes in these organisms may not be redundant, as shown previously in *E. histolytica* for neutral and acidic SMases. Interestingly, aSMase6 from *E. histolytica* shows almost no homology to other SMases from the same organisms or *T. vaginalis* (Appendix A, Panel A, green arrow). 

To further explore the differences between SMase sequences, examples of bacterial hemolysins, SMases from *B. cereus*, *B. turingensis*, *Dictyostelium discoideum*, *Acantamoeba castellanii*, and *Giardia intestinalis* were compared with *E. histolytica* and *T. vaginalis* SMases using Clustal omega sequence alignment and then using Alignment Viewer, which generated a 2D pairwise comparison map (Appendix A, Panel A). As shown, there are limited hubs of low-identity proteins in a wider analysis. No correlation was found between either free-living versus parasitic organisms or any of the two groups with bacterial enzymes. The observation that no true clustering was observed was further tested using UMPA analysis (Appendix A, Panel B). UMPA analysis confirmed that the clusters found have mixed enzymes from different organisms, indicating that the enzymes have low sequence homology, and no grouping per lifestyle can be identified. 

Next, we tested whether the enzymes may contain conserved sequence signatures when comparing a larger group of sequences. Appendix A, Panel A, shows the sequence comparison between a set of the sequences analyzed in Appendix A, with the upper panel in the MSA color scheme and the lower panel in the hydrophobicity color scheme. Again, all enzymes show conserved physicochemical properties, suggesting that, even though the sequences have limited sequence conservation, the functionality is conserved, and perhaps the protein folding is also conserved across species. 

Sequence motifs may uncover conserved features across species in SMases. These sequences were analyzed for motif discovery using GLAM2. In Appendix A, Panel B shows the best ten motifs, ranging from 30 to 50 amino acids, with a score ranging from 0.12 to 0.33 using all enzymes analyzed in Appendix A (Panel A). As can be seen, no true motif can be identified. Since no motif could be identified, a focus on bacteria, *E. histolytica*, and *T. vaginalis* was taken, and the dataset was analyzed again. In Appendix A, Panel C shows the motif discovery results using GLAM2, showing ten motifs of 24 to 50 amino acids with scores ranging from 0.08 to 0.23. Again, no highly scoring-motifs were found. To assess whether these motifs could be associated with either bacterial or protist organisms, the motifs shown in Appendix A, Panel C, were mapped against the protein sequence using the MAST tool, and the result is shown in Panel D. As shown, some low-scoring motifs’ arrangement and combinations are preferentially present in specific organisms. However, no true correlation could be drawn from this analysis, suggesting again that these enzymes show high sequence diversity.

### 3.2. Structural Features of SMases

Zhou and colleagues [42] have determined the crystal structure of the human aSMase, a Zn-dependent enzyme in its monomeric form, showing that the active site contains two zinc ions and a histidine-rich active site (metallo-phosphatase domain), where the two zinc ions are needed to activate a water molecule that renders the nucleophilic attack on the phosphodiester bond in the substrate. Also, Xiong and colleagues [43] have determined that this enzyme is a dimer and that the saposin domain helps orientate the substrate in the active site and is relevant in placing the glycosylated moieties towards the aqueous solvent. Xiong and colleagues [43] also suggest that the saposin domain is highly flexible and involved in the deformation of the membrane, allowing the metal centers of the enzyme to approach the substrate. Thus, with the available information regarding human aSMase and that the aSMase6 from *E. histolytica* is involved in plasma membrane repair, in this work, we focused on two main aspects of aSMase6: the metal ions bound to this enzyme and the role of Co^2+^ as a non-specific inhibitor [26] and the effect on the predicted structure of this enzyme. Also, the role of post-translational modifications on the overall protein structure was explored. 

The first approach was to determine the monomer and dimer model of aSMase6 compared to the aSMase human crystal structure. In Figure 1, Panel A shows the AlphaFold3 model of *E. histolytica* SMase6 (UniProt accession number C4LVV3) compared with the human acid SMase (PDB 5I81). The protein from *E. histolytica* was modeled without signal peptide or the predicted glycosylated residues (Appendix A shows the predicted topology, the presence of the signal peptide, and the glycosylation sites of all *E. histolytica* enzymes, including aSMase6), including the C-terminal end processing previously shown [26] and containing two Mg^+2^ ions SMase6 (structure in light blue, ions in green), which are the activating ions for this enzyme [26]. Human aSMase (light gray, PDB 5I81) shows an RMSD of 2.79 Å and a TM score (normalized by the length of the shortest structure) of 0.88971, indicating that the model of aSMase6 with the post-translational modifications shows a highly similar fold as the human enzyme. Appendix A shows that the dimeric form of the human aSMase shows a more extended folding due to an N-terminal saponin domain (Appendix A, Panel A, for the dimeric structure and Panel B for the monomer as a reference), which is absent in the *E. histolytica* aSMase6 (Appendix A, Panels C and D). However, the structure shows a more relaxed interaction when the enzyme has two Mg^2+^ ions bound (Appendix A, Panels C and D). 

A detailed examination of the aSMase6 aligned with the human aSMase shows that the AlphaFold3 model positions the Mg^2+^ ions overlapped with the Zn^2+^ ions in the human enzyme, as shown in the zoomed-in region in Figure 1, Panel A. With the data presented in Figure 1, AlphaFold3 training is able to predict the position of Mg^2+^ in aSMase6 in the same pocket as in human aSMase, which allows for further exploring the biochemical data gathered for this enzyme with a structural point of view. Previously, Ramírez-Montiel and colleagues [26] demonstrated that the recombinant aSMase6 enzyme is inhibited with Co^2+^ in a concentration lower than 1.0 mM. The data are consistent with the enzymes secreted via the parasite. To further assess the possible role of ion binding on overall structure prediction, models of the dimeric form of the aSMase6 lacking the signal peptide and C-terminal end region were modeled in the presence of Mg^2+^ or Co^2+^ ions. The activity is also affected by a chelating agent (EGTA), which suggests that the metal ion bound to the enzyme is needed for activity. The structure resists the chelating effect since the activity is stable at 50%, even at a 20 mM concentration of EGTA [26]. In Figure 1, Panel B, two views of the SMase6 protein modeled as a dimer were aligned and show that the two proteins have strong variations between them, evidenced by an RMSD value of the alignment of 4.36 Å and a TM score of 0.82365 normalized by the length of the shorter structure, which is unexpected since the two proteins are identical except for the metal ion bound (indicated on Figure 1, Panel B, right image). The result shown in Figure 1, Panel B, suggests that the Co^2+^ ion may induce a different conformational state compared to the structure with the Mg^2+^ bound. Overall, the AplhaFold3 models are consistent with the structure of human aSMase, and the position of the ions bound to the enzyme suggests a conserved catalytic mechanism, as shown in the human enzyme. 

### 3.3. C-terminal Processing Putative Participants

Previously, Ramírez-Montiel and colleagues [26] demonstrated that the aSMase6 from *E. histolytica* is processed in the C-terminal end using fusion to a red fluorescent protein. The finding agrees with the processing shown in human aSMase; the C-terminal cysteines are subjected to proteolytic processing [44] and are dependent on cathepsin D [45]. The search for homologous proteins to cathepsin D in *E. histolytica* showed no highly homologous proteins. Still, three low-identity cysteine proteases suggested that they may be related to the C-terminal processing of aSMase6 during membrane trafficking before secretion. In Figure 2, Panel A, low-homology cysteine proteases (cysteine protease 1 [UniProt Q01957], cysteine protease 2 [UniProt Q01958], and cysteine protease 3 [UniProt P36184]) to cathepsin D were identified, and the modeled structures were aligned. The alignment of the cysteine proteases and cathepsin D showed an RMSD of 4.57 and a TM score of 0.59142 normalized by the length of the shorter structure. Using the UCSF Chimera software v.1.18, Clustal omega alignment shows conserved residues in the four structures (Figure 2, Panel A, shown in magenta). To estimate whether a join model of cathepsin D and aSMase6 from *E. histolytica* could position the C-terminal residues in a bond distance with the active site residue of cathepsin D, the AlphaFold3 model was generated. In Figure 2, Panel B, the AlphaFold3 model using human cathepsin D suggests that the interaction with aSMase6 agrees with the orientation and position of the key residues of cathepsin D (in green) for C-terminal end processing. In Figure 2, Panel C, the left panel shows the interacting residues between cathepsin D and aSMase6; in red, the aSMase6 C-terminal cysteine residues are shown, and in green, the interacting and catalytic residues of cathepsin D (where Asp^31^ is the key processing residue [46]) are shown. The right panel is the rotated image of the region where the key cathepsin D residues are located to the right, and measurements were taken using PyMOL. In all instances, the distances between the C-terminal cysteines of aSMase6 and cathepsin D range from 12.7 to 18.2 Å, even though the cognate processing enzyme is not human cathepsin D. To assess the effect of an active form of the enzyme, aSMase6 was modeled with two Mg^2+^ ions and in the presence of cathepsin D. In Figure 2, Panel D shows that the catalytic residues of cathepsin D are facing in the opposite direction of the C-terminal end of aSMase6, suggesting that the processing of the enzyme occurs when the enzyme is in its inactive state during vesicular trafficking, as shown for human aSMase [44]. The monomer-to-monomer interaction models were conducted based on the data shown in Figure 1 since the C-terminal domain is buried in the dimer interphase, as shown for the human PDB structure. 

The models of aSMase6 with the three candidate proteases were generated using the same approach. In Figure 3, the thre cysteine proteases shwon in Figure 2, panel A, were AlphaFold3 models interacting with SMase6, and each of the cysteine proteases was used. In Panel A, the model of aSMase6 (in purple) with cysteine protease 1 (UniProt Q01957, cyan) is shown. The predicted catalytic residue (via ScanProsite, Glu^69^) is shown. A closeup image to the right shows the position of the predicted catalytic residue, and measurements were taken, showing a distance of 3 and 3.4 Å with Cys^398^ and Cys^402^, respectively. This model suggests that the enzyme is the best candidate to evaluate the processing of aSMase6 since, as shown in Panel B and Panel C, cysteine proteases 2 and 3, the predicted catalytic residues are located far from the C-terminal end cysteines of aSMase6. To complement the value of these models, in Appendix A, the models of aSMase4 with cathepsin D, cysteine protease 1, cysteine protease 2, and cysteine protease 3, respectively, are shown; in all instances, none of these enzymes show structural features that suggest that these enzymes may process the C-terminal end of aSMase4. 

### 3.4. Structure Prediction with Post-Translational Modifications

Human aSMase has been shown to present glycosylation, and a key regulatory mechanism is phosphorylation in Ser^508^ mediated via protein kinase C [47,48]. 

No unique phosphorylating residue has been identified in aSMase6. Using NetPhos 3.1, 33 putative phosphorylation sites were identified (Appendix A). Considering that the key phosphorylation site (if any) is required to regulate the enzyme activity, we modeled the protein with all the predicted phosphorylation sites. 

In Figure 4, we address the structural variations predicted via AlphaFold3 when the protein is either glycosylated, phosphorylated, or both. First, does structural variation depend on the presence of signal peptide and/or the C-terminal end? To answer this question, in Figure 4, Panel A, a protein model comparison between structures without either a signal peptide or a C-terminal end is shown. In red, the full-length protein is shown; in green, no C-terminal end was included (processed at residue Cys^398^ as an approximate point of processing based on the models presented in the previous section and with the experimental data reported by [26]), and in blue, the protein without a signal peptide is shown. The RMSD of alignment is 0.98, and the length of the shorter structure normalizes the TM score of 0.99258. Now, is there a strong variation if the protein structure is glycosylated? To explore this possibility, in Figure 4, Panel B, the aSMase6 protein lacking the signal peptide and C-terminal end was modeled with the three predicted glycosylation sites and compared when no ion was present (shown in red), and with Mg^2+^ (in green) and Co^2+^ (in purple). The RMSD for the alignment is 1.16, and the TM score is 0.98489, normalized by the shorter structure’s length. The structural variation suggests that the metal bound influences the three glycosylation moieties’ positions. The arrows in the figure show the positions where more variation is seen through inspection by eye, suggesting that the upper loop may be highly flexible and exhibit a nov role, as the saposin domain was shown for the human aSMase [42]. 

Finally, what structural and regulatory roles may the phosphorylation of aSMase6 play? In Figure 4, Panel C, the models now include the predicted phosphorylation sites (in red) and glycans (in blue) when modeled with Mg^2+^ (in green) or Co^2+^ (in purple) ions. Here, the protein structures show a transition of a compact loop (dashed arrow in the model without ions). The loop is bent towards the center of the structure when two ions are modeled. In the case of Mg^2+^, one of the ions is predicted to be farther from the center of the structure, which strongly differs from the model shown in Figure 1. The rest of the structure does not vary, depending on the predicted modeled post-translational modification included (RMSD of 1.42 Å, and TM score normalized by the length of the shorter sequences of 0.93918). The structural transitions suggest that the loop (residues 308-334) could regulate the accession to the substrate and may be a highly dynamic protein region. Also, this may regulate the selectivity and binding of the activating ion. 

### 3.5. Substrate Docking

High-quality protein models have become novel tools for assessing molecular docking with natural and inhibitory molecules on pharmaceutically relevant targets [49]. Taking advantage of fast and accurate tools for molecular docking, such as CB-Dock2, which allows the detection of cavities in a protein and blindly assesses the binding of sphingomyelin, the models generated in this work were prepared for molecular docking; Vina score (the weighted sum of atomic interactions [50] in kcal/mol) and cavity size (in Å) were used to assess the binding of the substrate, along with the predicted resulting model. In the case of the heavily phosphorylated protein models, no docking could be generated due to stereochemical clashes between the substrate and the phosphate moieties. 

The cavity-detection blind docking analysis of the aSMase6 protein containing post-translational modifications was conducted. In Figure 5, we show the best-predicted region containing the bound substrate. In Table 1, the Vina scores and cavity size are reported. In all instances, the best-scoring cavity and substrate binding was found in the same region as shown in Panel A, where the location of the substrate-bound is shown for a C_9_H_21_N_2_O_6_P sphingomyelin (CID number 70678679, using the 3D conformer) in a protein model lacking the signal peptide and if an ion is bound. A phosphorylated protein model could not be used due to strong stereochemical clashes in the structure that prevented any predicted binding. However, all the structural variations, such as lacking the N-terminal signal peptide or the C-terminal end processing, were analyzed for substrate binding in the absence of different ions. In Figure 5, Panel A, from left to right are shows the full protein as a reference, showing the surface colored for the Coulombic charge (blue for negative, red for positive, and white for hydrophobic), protein with no ion, protein with Mg^2+^ bound, and protein with Co^2+^ bound. The substrate color scheme is for heteroatoms with carbon and hydrogen in white, nitrogen in blue, and oxygen in orange. Here, the substrate is predicted to bind in two special configurations, one in the protein without ions, the same when Co^2+^ is bound, and another when Mg^2+^ is bound to the protein. Then, the fully processed protein containing glycosylated residues was tested for docking with the indicated ions, and the results are shown in Figure 5, Panel B. Here, as shown experimentally, the position of the substrate is different in the Mg^2+^ bound protein but not in the rest of the predicted models. Here, the position of the substrate is influenced by the glycosylation and the metal ions, showing different positions for Co^2+^ and Zn^2+^ in comparison with the protein with no metal ion bound. The substrate binding is modified if the protein is modeled with two ions, as shown in Figure 5, Panel C. The docking suggests that the substrate is reversed when Co^2+^ is bound, and this is regardless of the N-terminal domain since, as shown in Panel D, aSMase6 was modeled with the previously reported 6xHis tag for recombinant protein production [26] (residues added from the pRSET A plasmid) and full-length C-terminal end containing two Mg^2+^ or Co^2+^ ions, showing that the substrate was predicted to occupy the same position as shown in Panel C and is in agreement with the biochemical data reported previously [26]. Additionally, in all the predictions, when Co^2+^ was modeled in the protein, the substrates shifted outwards from 6.1 to 18.2 Å from the end of the cavity, which is seen in the figure as a shift to the right on each model. Overall, the docking data suggest that the Co^2+^ inhibitory mechanism is exerted by placing the substrate farther from the active site; further analysis via scanning mutagenesis could shed light on the residues involved in the coordination of Co^2+^ in aSMase6.

The limitation of using CB-Dock2 lies in the Vina scoring function, a weighted sum of atomic interactions. The models used in this work include charged glycan moieties, and their steric, hydrophobic, and hydrogen bonding interactions were calculated. To account for entropic penalties, the Vina score was reduced when the interactions were adjusted based on the number of rotatable bonds in the sphingomyelin substrate [50]. The data, however, are consistent with the metal-bound cavities, and no other cavities were detected in the protein, providing confident information to target key residues for mapping the active site in these enzymes. 

**Table 1 pathogens-14-00032-t001:** Cavity detection-guided blind docking score features for SMase 6 with different combinations of processing and post-translational modifications.

Protein Features	Vina Score (kcal/mol)	Cavity Volume (Å^3^)
SMase 6, no signal peptide	−5.6	286
SMase 6, no signal peptide +Mg^2+^	−4.9	209
SMase 6, no signal peptide +Co^2+^	−5.1	219
SMase 6, no signal peptide, no C-terminal end, glycosylated	−5.1	299
SMase 6, no signal peptide, no C-terminal end, glycosylated +Mg^2+^	−4.9	380
SMase 6, no signal peptide, no C-terminal end, glycosylated +Co^2+^	−5.1	271
SMase 6, no signal peptide, no C-terminal end, glycosylated +Zn^2+^	−4.9	303
SMase 6, no signal peptide, no C-terminal end, glycosylated +2XMg^2+^	−5.2	294
SMase 6, no signal peptide, no C-terminal end, glycosylated +2XCo^2+^	−4.3	150
SMase 6, no signal peptide, No C-terminal end, no loop, glycosylated	−5.0	379
SMase 6, no signal peptide, No C-terminal end, no loop, glycosylated +2XMg^2+^	−5.2	230
SMase 6, no signal peptide, No C-terminal end, no loop, glycosylated +2XCo^2+^	−5.1	249

### 3.6. Evolutionary Analysis of SMases

The structural data shown in the previous sections suggest that the SMases are conserved enzymes. However, no evident conserved regions or motifs were found at the sequence level. 

To assess the divergence that SMases could exhibit, an evolutionary history analysis was conducted using enzymes from the two parasitic protists, *E. histolytica* and *T. vaginalis*. The analysis included other sources, such as free-living protists and bacteria producing hemolysins with SMase activity. In Figure 6, Panel A, the evolutionary history of all the SMases from *E. histolytica* and *T. vaginalis* shows that there are two groups linked to a common ancestor that is also related to *E. histolytica* UniProt accession number C4M5N7. The data presented in Figure 6, Panel A, suggest these enzymes are conserved in parasitic protists. 

To confirm this idea, an evolutionary history analysis was conducted, using sequences from *B. cereus* as a bacterial example and producing SMases with hemolytic activity, with two free-living protists (*Acanthamoeba castellani* and *Dictyostelium discoideum*) and additionally to the main subject organisms of this manuscript, *Giardia intestinalis*. The results are shown in Figure 6, Panel B. The evolutionary history of these enzymes suggests that the most diversified group is the bacterial hemolysins, while the eukaryotic enzymes remain closer in evolutionary terms. The phylogenetic distribution does not reflect a relationship between environment and lifestyle among the evaluated examples since no clear clustering is observed in the organisms sharing the same lifestyle or environmental conditions, for example, *G. intestinalis* and *E. histolytica*. This was further supported when the phylogenetic relationships evaluated included five SMases from humans (Appendix A), or the comparison was only performed with *D. discoideum*, *A. castellani*, *E. histolytica*, and *T. vaginalis* (Appendix A). The evolutionary history analysis, therefore, suggests that SMases are highly divergent proteins in sequence identity but preserve evolutionary relationships that could imply structural conservation. 

Based on the phylogenetic relationships among SMases that suggest some sequence-level conservation between them, the focus was on hemolysin proteins from bacteria to uncover hints linking these enzymes to hemolytic activity in the SMases from protozoan organisms. In Figure 7, Panel A, using hemolysin enzymes from different pathogenic bacteria, six aSMases from *E. histolytica* are phylogenetically distant from the enzymes analyzed, except for aSMase C4M8T3 (aSMase5), which shows strong evolutionary relationship with one of the S. aureus enzymes, and aSMase C4LYZ5 (aSMase3), which shows a linkage to Leptospira interrogans hemolysins. The previously characterized aSMase6 was found to be linked to aSMase A0A175JZ26 (aSMase2) and aSMase4. Overall, the topology of phylogenetic analysis suggests that SMase enzymes may have common structural features since they have no sequence homology but have a shared evolutionary history. To test this, structural models retrieved from UniProt were aligned to correspond to each highlighted protein analyzed in the phylogenetic tree. In Figure 7, Panel B, the comparison of all the enzymes shows a common structural core, apart from aSMase6 of *E. histolytica* loop near the C-terminal end (dashed arrow), which also contains conserved residues with the hemolysins (lower alignment, residues indicated in magenta). In this case, the RMSD is 3.03 Å, and the TM score is 0.74416, normalized by the length of the shorter structure. This loop is the only region different from the six acid SMases of *E. histolytica* (Figure 7, Panel C, RMSD of 2.41Å and a TM- score of 0.86749 normalized by the length of the shorter structure). 

To test this, recombinant aSMase4 and aSMase6 proteins were produced in *E. coli*, purified, and tested for their hemolytic activity using human red blood cells at pH 7.0 (neutral) and pH 5.0, which is the optimal pH for acidic SMases. The results shown in Figure 7 Panel C indicate that, at pH 7.0, the control cells do not release hemoglobin (row 1). As positive control, *B. cereus* hemolysin was used (from a commercial source, Sigma-Aldrich, row 2), which is active at pH 7.0 but not at an acidic pH. In the case of aSMase4 (in row 3), slight hemoglobin releases were observed at pH 7.0, and the discrete hemolytic activity observed at pH 5.0 is much lower than for aSMase6 (row 4), which has strong hemolytic activity. The most notorious feature that differs among the structures is the loop in aSMase6, which is absent from the rest of the aSMases from *E. histolytica*. 

Further analysis of aSMase4 indicated that the protein has more flexibility in the C-terminal end (Figure 8, Panel B, and for comparison, in Panel A the structure of aSMase6) when modeled with glycosylation and ions bound (Mg^2+^ or Co^2+^). In the case of aSMase6, when modeled in the absence of the loop that is not present in the other SMases, the protein remains with the same overall structure (Figure 8, Panel C). The predicted binding site for sphingomyelin is the same, even in the presence of Co^2+^, which suggests that the loss of this loop may result in resistance to inhibition for this ion. The latter hypothesis remains to be elucidated experimentally. 

### 3.7. T. vaginalis Enzymes Show Structural Divergence from the E. histolytica Counterpart

The structural analysis of all *T. vaginalis* enzymes revealed something unexpected, except for one (XP_001296506.2); the enzymes are predicted to be membrane-bound via an N-terminal extension that contains an α-helix (Figure 9, Panel A). Protter analysis failed to identify a signal peptide sequence in the analyzed enzymes (Figure 9, Panel A). The membrane-bound predicted helix could belong to a non-canonical signal peptide. However, the comparison of all the putative membrane-bound domain sequences failed to identify a conserved signature that could be consistent with the processing of a signal peptide (Figure 9, Panel B), suggesting that either the domain is needed to bind the enzymes to the membrane or is a non-identified signal peptide, which requires further scrutiny. Nevertheless, in *T. vaginalis*, the SMases have a completely different topology than in *E. histolytica*. 

The analysis of structural models’ quality in a predicted local distance difference test (pLDDT) in *T. vaginalis* SMases showed high quality in the protein structure (Appendix A) with a lower score on post-translational modifications, especially when attempting to confirm the validity of the transmembrane domain by using 50 oleic acid subunits included in the model in a rough attempt to emulate a membrane domain. The metal binding pocket (and, presumably, the active site) in the membrane-bound enzymes was either located towards the membrane, according to the prediction of the membrane-bound domain, or in the opposite direction (Appendix A with an active site towards the membrane, while Appendix A indicate the opposite orientation). Enzymes A2EE69 and A2E5Q5 (Appendix A) showed that the transmembrane domain changed radically when ions were included in the model. The quality of the model dropped at this helix. It was recovered when oleic acid was included in the model, suggesting that the membrane domain in these enzymes may be more flexible and could regulate the accession of the enzyme to the substrate. The rest of the enzymes did not display this transition in the membrane domain. Further experimental scrutiny of these enzymes is needed to assess their true association with the cell membrane. 

It is crucial to continue to validate our findings through experimental scrutiny. The only predicted soluble enzyme (XP_001296506, Appendix A) showed a structured loop (the last 30 residues) in a similar position as the loop in aSMase6 in *E. histolytica* but showing a partially structured α-helix and the sequence contains two cysteine residues, which could be processed as suggested in previous lines. The quality of the model is not altered when Mg^2+^ or Co^2+^ ions are included in the model, suggesting that the enzyme inhibition through Co^2+^ remains to be elucidated experimentally. 

With all the above, the evolutionary origin of these enzymes remains to be elucidated. Cordes and Binford [52] uncovered the evolutionary origin of dermonecrotic toxin, a unique SMase present in the venom of spiders from the genus Loxosceles. It is also present in certain pathogenic bacteria, such as Corynebacteria, suggesting an ancestral origin in bacteria and then lateral gene transfer from one lineage to the other. The authors identified the glycerophosphodiester phosphodiesterase enzymes as a long-standing ancestor. 

In the previous sections, the phylogenetic analyses suggest, as Cordes and Binford [52] found, that SMase enzymes may have a more conserved origin than previously thought. In Figure 10, Panel A, using an evolutionary history analysis with all the enzymes from *E. histolytica* and *T. vaginalis*, including the dermonecrotic toxin of the spider *Loxoceles laeta* and the putative ancient glycerophosphodiester phosphodiesterase from *E. coli*, clearly separates the acid SMases of *E. histolytica* and a group of SMases of *T. vaginalis* and groups with a common ancestor with the dermonecrotic spider toxin. Meanwhile, the other group of enzymes from both protist groups shows the putative ancient enzyme from *E. coli*. The analysis suggests that SMases may share a common core element, as Cordes and Binford [52] suggested.

To test the hypothesis that SMases may contain a common core, the structural features of the available structures or AlphaFold3 models generated in this work were compared with the structure of the two SMase enzymes from spider and *E. coli*. In Figure 10, Panel B, are shown, on the left, the rainbow color scheme of the model from L. laeta SMase D, on the right, the structural alignment with human aSMase (PDB 5I81, in red) with *E. histolytica* glycosylated and with two Mg^2+^ ions nSmase6 (in blue), and in green, the *L. laeta* SMase D model. The alignment showed an RMSD of 4.49 Å and a TM score normalized by the length of the shorter structure of 0.58819. In Panel C, the structural alignment of the *L. laeta* SMase D (in green) and aSMase6 (glycosylated and with two Mg^+2^ ions) of *E. histolytica* enzyme (in blue), showing an RMSD of 5.37 and a TM score normalized by the length of the shorter sequence of 0.46147, is shown. The structural alignments show that, even though the RMSD and TM scores are not ideal, these proteins have features conserved, especially the helixes surrounding the active site and the β-sheets in the back of the active site. In Figure 10, Panel D, a comparison between the crystal structure of *E. coli* glyercerophosphodiester phosphodiesterase (PDB 1YDY, in green) with the predicted dimer of *E. histolytica* aSMase6 (glycosylated and with two Mg^+2^ ions) is presented, showing an RMSD of 7.84 and a TM score normalized by the length of the shorter structure of 0.25975. This result is consistent with a more distant enzyme, but as shown, the general structure of the protein is conserved. Overall, the data in Figure 10 are consistent with an ancestral link with minimal SMase (spider toxin), suggesting that the functional fold is conserved, or with even more divergent protein than the glyercerophosphodiester phosphodiesterase, which is also consistent with a conserved fold.

## 4. Discussion

The avenue that the recent Nobel prize awarded to research using AI for protein prediction, AlphaFold2, and the novel AlphaFold3, opened the opportunity to explore the structure–function relationships in proteins, now specifically including post-translational modifications that are key in understanding their regulatory roles in proteins. In this work, using the predictive features of AlphaFold3 to integrate ligands and post-translational modifications in proteins, the structural variations due to different metal ions and post-translational modifications in acid sphingomyelinases with a focus on two parasites, *E. histolytica* and *T. vaginalis*, were assessed. Also, an evolutionary history analysis was conducted using examples of proteins from organisms with different lifestyles.

The results presented in this manuscript summarize the following.

First, AlphaFold3 models of *E. histolytica* aSMase6 are consistent with the structure of human aSMase and fit well with the position of Zn^2+^ ions needed for activity, determined via protein crystallography. The same analysis conducted with the enzymes from *T. vaginalis* reveals a completely different scenario; most enzymes are predicted to be membrane-bound, and the orientation of the active site is either towards the membrane or in the opposite direction. 

Second, the models showed something unexpected; since the exposure of aSMase6 from *E. histolytica* to Co^2+^ results in the total inhibition of the activity, even at low concentrations, the obvious prediction would be important structural transitions when predicted to be bound to this ion as a mechanism of enzyme inhibition. However, the models suggest structural variations without observing major changes in the overall structure, regardless of whether the C-terminal end is processed or not. The protein involved in C-terminal processing could be a cysteine protease, as suggested here. The predicted binding of the substrate suggests that the mechanism of Co^2+^ inhibition occurs by placing the substrate ~12 Å farther than the predicted binding site using Mg^2+^ ions. 

Third, the evolutionary relationship between enzymes points to a conserved overall fold and structure, rather than at the sequence level. Importantly, the biggest difference between the *E. histolytica* aSMases was a loop near the C-terminal end. Also, the loop may be associated with the hemolytic activity demonstrated for aSMase6, which aSMase4 lacks; it shows a different dynamic when either Mg^2+^ or Co^2+^ ions are predicted in the structure. 

Finally, the proposal by Cordes and Binford [53] is well supported in evolutionary terms, among not only the bacterial ancestors for SMase D found in spider toxins but also all SMases analyzed in this work. This suggests that SMases share a common ancestry, that the core of the enzyme is highly conserved, and that the evolution of additional sequences rendered enzymes with specialized functions. The use of good-quality protein models, such as the ones presented in this manuscript, can lead to the search for more homologs using the tool FoldSeek [54] or similar tools, which can allow the identification of low-sequence homology proteins but with conserved folds, as shown here. 

The most characterized enzymes in humans are acid SMases due to their relationship with diseases, specifically, the mutations or variants that affect their stability and accessibility to the substrate and the severity of the diseases (Niemann–Pick disease) [53]. Scrima and colleagues [53] show that the analysis of SMases using bioinformatics is limited due to the lack of tools for predicting post-translational modifications, which AlphaFold3 now bypasses. The greatest limitation of AlphaFold3 is that the predictions are not great (low pLDDT). However, they are a good approximation for designing future experiments, focusing on consistent models and the regions with the most disorder or low pLDDT values. The collection of identified mutants by Scrima and colleagues [53] and the data presented here open the opportunity to explore in more detail the relevant residues for activity in human aSMases and other biological systems (pathogenic microorganisms, cancer, metabolic disorders, etc.) since these enzymes are structurally conserved. 

The role of Zn^2+^ in the human aSMase has been extensively studied and forms a trigonal bipyramidal coordination geometry [42]. Here, the models were generated using the *E. histolytica* enzyme, which is Mg^2+^-dependent [26], suggesting an identical binding mode as described for the human enzyme. The Mg-dependent enzyme from *E. histolytica* is not a special case in humans since the neutral SMase that needs Mg^2+^ has been shown to be brain-specific [55]. Key catalytic differences between SMases could shed light on the mechanism of substrate binding and hydrolysis between the different enzymes that require different metal ions for activity. 

Human aSMase has been shown to present glycosylation and a key regulatory mechanism of phosphorylation in Ser^508^ mediated via protein kinase C [47,48]. In this sense, protein phosphorylation is key for *E. histolytica* virulence since Anwar and Gourinath [56] identify 250 non-redundant phosphatases in *E. histolytica*, which is 3.1% of the total proteome of this organism, suggesting an important role for this post-translational modification. The finding on the metal ion bound and the changes in the near C-terminal loop of aSMase6 suggests that searching for the phosphorylated residues in this enzyme may uncover its regulation and role in host invasion and, perhaps, a novel target for inhibition. The role of aSMase6 as a key element of membrane repair in *E. histolytica* [26] creates a potential candidate as a target for treatment. 

The structural models presented here are in concordance with the data generated in Leptospiral sphingomyelinase, where the enzymes are Mg-dependent and exhibit strong conservation with the bacterial hemolysins [57]. As shown for *L. ivanovii* and *B. cereus* enzymes that share a high-level structural similarity and the *L. ivanovii* enzyme, as shown here, both exhibit the SMase and hemolytic activity [57]. 

Even though SMase enzymes do not share sequence similarity or identity, structurally, as shown here, they are conserved. A strong case is presented via the transfer from bacteria to spider toxins, as shown by Cordes and Binford [52], which is relevant since these enzymes are conserved and functionally related. As shown here, the overall fold is essentially conserved. The results presented here suggest that the conservation of SMases at the structural level shows high conservation, and in the case of *E. histolytica*, the aSMase6 enzyme points towards a bifunctional enzyme, dependent on the reaction environment and, perhaps, a molecular target for inhibition. Due to its relevant enzymatic activity, the inhibition may ultimately result in cell death due to the lack of membrane repair activity and the inhibition of hemolysis, and possibly, other biological activities associated with colonization since SMases and the production of ceramide regulate other cellular processes such as endosome formation to repair the lesion and vesicle mobilization [23,58,59], which, in the case of pathogenic parasites, may also be associated with the secretion of virulence factors. Overall, SMases are conserved and highly important in eukaryotic organisms in membrane damage repair and cell survival, coupled with signal transduction and cellular response [60]; additionally, SMases are found to play distinct roles, such as toxins in bacteria and spiders. The evolution of these enzymes began early in life and then evolved to a myriad of cellular roles. 

Overall, the genetic diversity found in parasitic protists points towards complicated evolutionary pathways leading to the establishment of non-redundant networks of proteins involved in several physiological processes. Lessons learned from the diplonemids and euglenids indicate that the parasitic kinetoplastids sustained several metabolic losses, and genetically speaking, they lost most of the introns in their genomes [61]. Butenko and colleagues [61] pose a very interesting question: Could the ancestors of the kinetoplastids be a symbiont? The origin of the obligate parasites could extend to all the parasitic protists that share the same features as described by Butenko and colleagues, suggesting that the genome streamlining happened in free-living kinetoplastids. The data presented here suggest that the SMases do not have any clear signature associated with the lifestyle of the bearing organism; also, the data suggest a different view of the evolution of parasite–host interactions. Luong and Mathot [62] proposed that the intermediary step between free-living and parasitic lifestyles may be involved in the facultative parasitic lifestyle, which implies a ‘pre-adaptation’ through which the parasite gains more advantages while living inside a host. The view presented here suggests the following: the presence of enzymes with two greatly dissimilar activities, such as membrane repair and hemolytic activity, supports this notion and further advances our understanding of the parasite–host interaction in the sense that enzymes with limited sequence conservation but the exhibition of a conserved fold can bypass the membrane damage, provide nourishment to the cell, and allow reproduction. The notion of the adaptation of parasites to hosts is further supported by the presence of numerous genes encoding different physiological functions in the parasites *E. histolytica* and *T. vaginalis*, for example, Rab GTPases [63], for which only a few Rab GTPases have been associated with physiological processes [64], but it can be anticipated that they are not redundant. Several gene duplications could have arisen from stress elicited by the host or environmental conditions, as shown by other organisms such as *Saccharomyces cerevisiae* [65], in which common [66] and noncommon environmental stressors render genomic alterations and adaptability in this organism, such as interspecies hybrid formation or genomic landscape modifications resulting in adaptability [67,68]. Since the case of SMases does not show strong evolutionary divergence in terms of folding (only in sequence), experimentally, a strong regulatory ion dependence is observed, and thus, functional divergence is observed between toxins and enzymes associated with membrane repair; this may concord with multiple attempts to establish a survival toolbox, rather than an ‘aggressive’ toolbox, and it could be associated with an early whole-genome duplication event that led to the presence of multiple copies of these genes, which has been documented [69], resulting in the divergence of the sequence and function in *E. histolytica*. 

Studying different pathogenic microorganisms allows the discovery of novel virulence factors and determinants. The data presented here suggest that the SMases in *T. vaginalis* have unique features, such as the majority containing a predicted membrane-bound domain, and at least two orientations of the enzyme are predicted. Recently, the role of exosomes and microvesicles in *T. vaginalis* has been associated with host attachment, immunomodulation, and virulence factors cargo for exosomes and host–parasite interaction and parasite–parasite interaction, as well as virulence factors’ cargo for microvesicles [70]. However, the analysis of exosomes’ cargo did not show any SMases in their content in the study by Twu and colleagues [71], but three enzymes (UniProt A2F5C2 [TVAG_222460], A2E5Q [TVAG_271580], and A2G7J3 [TVAG_020780 or XP_001298660.1]) were identified by Ong and colleagues [72] since the enzymes found contain a putative membrane domain; how are these enzymes bound to the exosome, and how do they exert their activity? Further scrutiny will be needed to assess whether any SMase is secreted or whether they are located in the vesicle membrane, which can be explored by stimulating the formation and release of microvesicles with Ca^2+^ [73]. Overall, we hypothesize that the action mechanism of these enzymes differs, depending on the organism and the environment in which they exert their function. 

Following the previous line of thought, membrane-associated domains in the SMase enzymes in *T. vaginalis* suggest a novel secretion signal in this organism. Previously, Padilla-Vaca and Anaya-Velazquez [74] reported a secreted activity of a membrane-associated neuraminidase enzyme in *T. vaginalis*. Perhaps the only possibility for the secretion of the *T. vaginalis* enzymes containing a putative signal or targeting sequence could be as reported by Eberle and colleagues [75], where the endocytosis of lysosomal acid phosphatase requires a Tyr-containing signal in the cytoplasmic domain and is present in an 18-amino acid peptide [75], which may work inversely in *T. vaginalis* or be associated with its secretion in exosome-like vesicles. 

Finally, the role of SMases in pathogenic anaerobic protists may also be associated with extracellular vesicle formation and loading, which is a new and exciting field of research [70] that is relevant to host invasion, immunomodulation, and virulence factors in cargo deployment. It may also be associated with parasite–parasite interaction [70]. In the case of *T. vaginalis*, the topology of the active site may be associated with the membrane it shapes, suggesting that the enzymes with the active site placed towards the cell membrane may regulate the rate of exosome and microvesicle formation, their release from the cell, and perhaps their fusing to the target cell. 

## 5. Conclusions

In conclusion, the results presented in this manuscript point towards the more extensive relevance of experimental structural analyses of protozoan SMases and strongly suggest no redundant activity (via either intracellular localization or compartment activation) in both *E. histolytica* and *T. vaginalis*. These enzymes are good candidates for novel therapeutic targets. Further experimental analysis is needed to assess their role during cell survival and host invasion.

## Figures and Tables

**Figure 1 pathogens-14-00032-f001:**
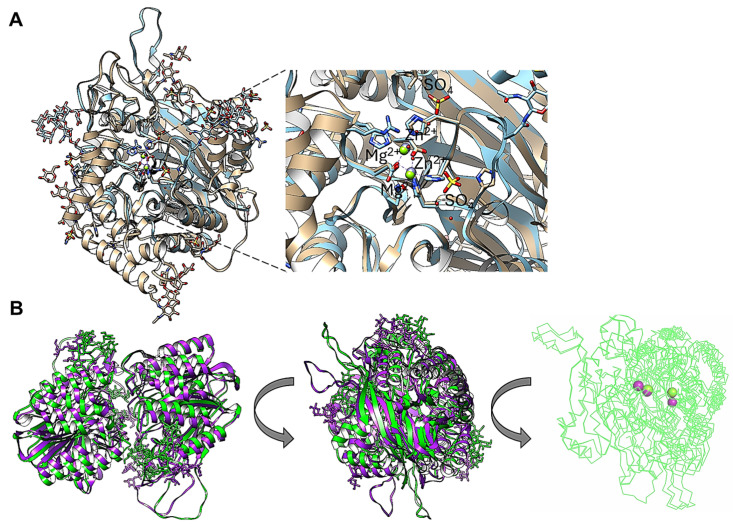
The AlphaFold3 model of *E. histolytica* SMase6 (UniProt accession number C4LVV3) shows high structural conservation against human acid SMase (PDB 5I81). In Panel (**A**), the US-align between *E. histolytica* is glycosylated and C-terminal end processed and containing 2 Mg^2+^ ions SMase6 (structure in light blue). Human aSMase (structure in light gray, PDB 5I81) shows an RMSD of 2.79 Å and a TM score (normalized by the length of the shortest structure) of 0.88971. The zoomed-in region shows the exact overlap of the Mg^2+^ ions (in green) and Zn^2+^ ions (in gray); sulfate ions from the human aSMase crystal are indicated. Panel (**B**) shows two views of the SMase6 protein models as a dimer alignment with US-align. The protein structure with two Mg^2+^ is shown in green, and the two Co^2+^ ions are shown in purple. The RMSD value of the alignment of the two models is 4.36 Å and a TM score of 0.82365 normalized by the length of the shorter structure. To make the metal ions visible, the position shown in the middle was modified to a ribbon structure visualization, and the metal ions are colored in green for Mg^2+^ and purple for Co^2+^ and slightly tilted to the left so that the two ions are visible.

**Figure 2 pathogens-14-00032-f002:**
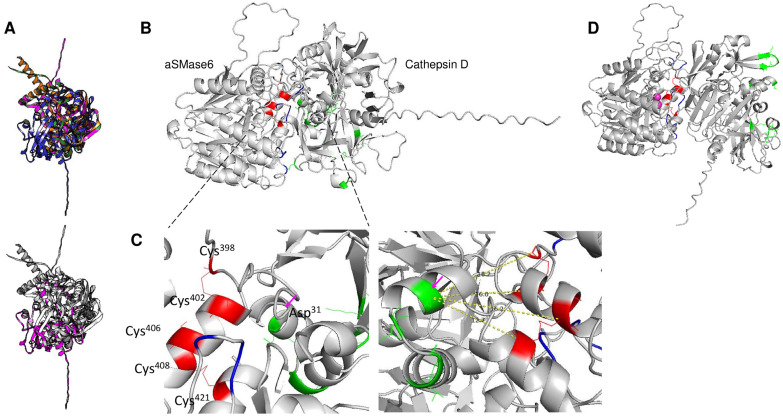
Mammalian aSMases require C-terminal end processing. aSMases require the processing of the C-terminal end via cathepsin D, which was not found through BLAST in *E. histolytica*. Panel (**A**) low-identity cysteine proteases were identified and modeled, and the structure model’s alignment showed an RMSD of 4.57 and a TM score of 0.59142 normalized by the length of the shorter structure. In blue, the full-length human cathepsin model is shown; in green, cysteine protease 1 (UniProt Q01957) is shown; in magenta, cysteine protease 2 (UniProt Q01958) is shown; and in orange, cysteine protease 3 (UniProt P36184) is shown. Using the UCSF Chimera software, Clustal omega alignment shows conserved residues in the four structures (lower panel in magenta). Panel (**B**) AlphaFold3 model using human cathepsin D suggests that the interaction with aSMase6 agrees with the C-terminal end processing. In Panel (**C**), the left panel shows the interacting residues between cathepsin D and aSMase6; in red, the aSMase6 C-terminal cysteine residues are shown, and in green, the interacting and catalytic residues of cathepsin D (where Asp31 is the key processing residue) is shown. The right panel is a rotated image of the region where the key cathepsin D residues are located to the right, and measurements were taken using PyMOL. In all instances, the distances range from 12.7 to 18.2 Å. Panel (**D**) shows the model when the aSMase contains two Mg^+2^ ions in the presence of cathepsin D; the catalytic residues of cathepsin D are facing in the opposite direction of the C-terminal end of aSMase6.

**Figure 3 pathogens-14-00032-f003:**
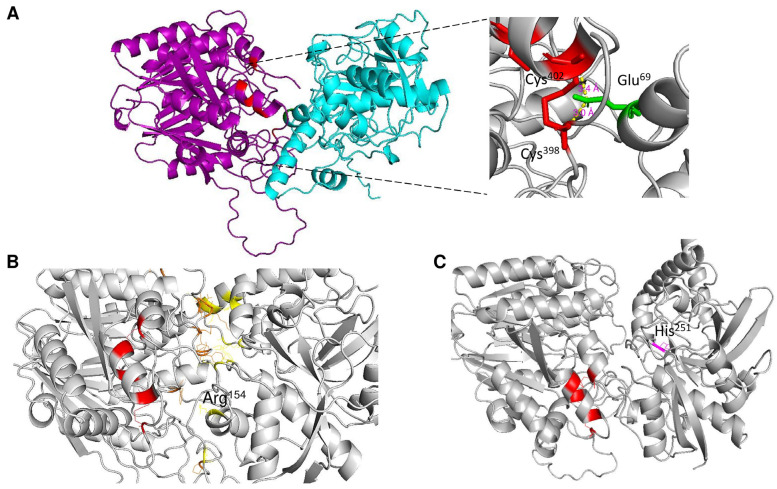
C-terminal end processing of aSMase6 may be a cysteine protease from *E. histolytica*. Using the three cysteine proteases shown in Figure 2, AlphaFold3 models using aSMase6 and each cysteine proteases were found to have low sequence homology, but relevant structural conservation with cathepsin D was used. In Panel (**A**), the model of aSMase6 (in purple) with cysteine protease 1 (UniProt Q01957, cyan) is shown. The catalytic residue (Glu^69^) is shown. Closeup images show the position of the predicted catalytic residue, and measurements were taken, showing a distance of 3 and 3.4 Å with Cys^398^ and Cys^402^, respectively. Panel (**B**) shows the model of aSMase6 with cysteine protease 2 (UniProt Q01958), showing the catalytic residue. As shown, the C-terminal end cysteines of aSMase6 are far from the catalytic residue. Panel (**C**) shows the model of aSMase6 with cysteine protease 3 (UniProt P36184). As shown, the C-terminal end cysteines of aSMase6 are far from the catalytic residue.

**Figure 4 pathogens-14-00032-f004:**
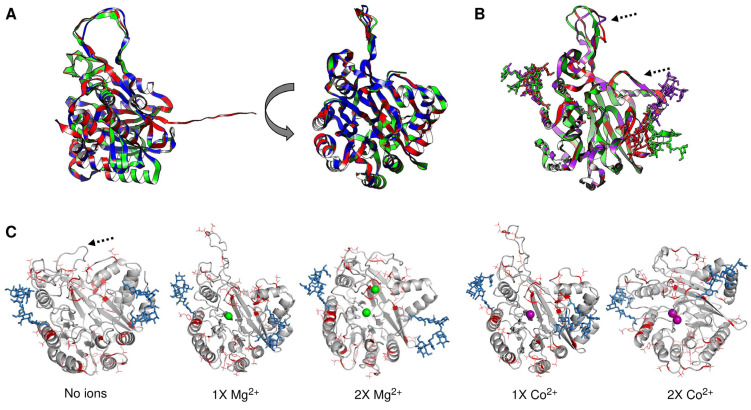
Structural effects of post-translational modifications on aSMase6. In Panel (**A**), protein processing analysis is shown by comparing structures without either signal peptide or C-terminal end. In red, full-length protein is shown; in green, no C-terminal end was included (processed at residue Cys^398^, and in blue, the protein without signal peptide). The RMSD of alignment is 0.98, and the length of the shorter structure normalizes the TM score of 0.99258. In Panel (**B**), the aSMase6 protein lacking the signal peptide and C-terminal end was modeled with the three predicted glycosylation sites and compared when bound without ion, with Mg^2+^ (in green) and Co^2+^ (in purple). The RMSD for the alignment is 1.16, and the TM score is 0.98489, normalized by the shorter structure’s length. Panel (**C**), structural features of aSMase6 when modeled, including the predicted phosphorylation sites (in red) and glycans (in blue) when modeled with Mg^2+^ (in green) or Co^2+^ (in purple) ions. The position of the proteins is the same as in the previous panels.

**Figure 5 pathogens-14-00032-f005:**
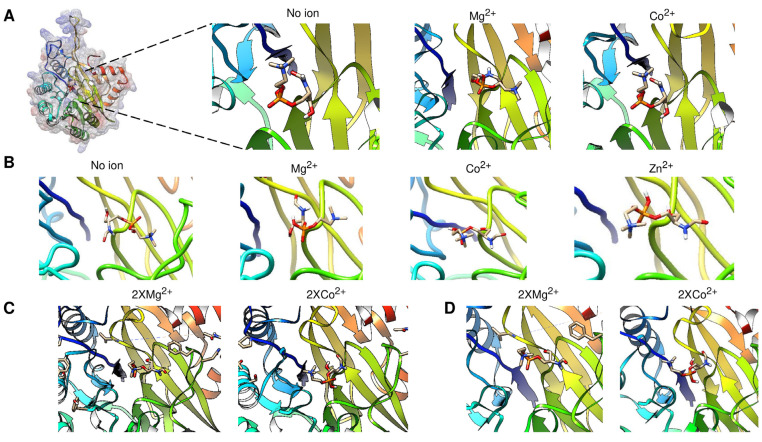
Cavity-detection blind docking analysis of the aSMase6 protein containing different structural variations. Panel (**A**) shows the location of the substrate (a C_9_H_21_N_2_O_6_P sphingomyelin, CID number 70678679, using the 3D conformer) in a protein model lacking the signal peptide and if an ion is bound. From left to right are shown full protein as a reference, showing the surface colored for Coulombic charge (blue for negative, red for positive, and white for hydrophobic), protein with no ion, protein with Mg^2+^ bound, and protein with Co^2+^ bound. The substrate color scheme is for heteroatoms with carbon and hydrogen in white, nitrogen in blue, and oxygen in orange. The fully processed protein and glycosylated, with the indicated ions, was used for docking in Panel (**B**). In Panel (**C**), the docking was performed in a protein model containing two Mg^2+^ or Co^2+^ ions. Panel (**D**) aSMase6 modeled with the previously reported 6xHis tag for recombinant protein production and full-length C-terminal end containing 2 Mg^2+^ or Co^2+^ ions. In all instances, when Co^2+^ is modeled in the protein, the substrates shift outwards from 6.1 to 18.2 Å from the end of the cavity.

**Figure 6 pathogens-14-00032-f006:**
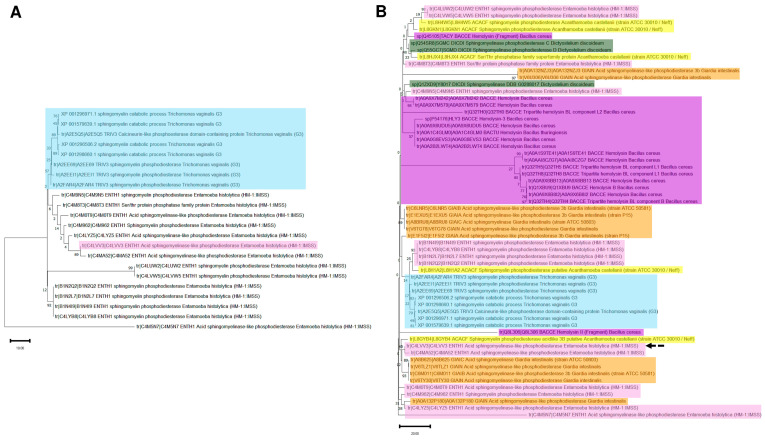
Evolutionary analysis of SMases from different microorganisms suggests closer relationships than previously anticipated. Panel (**A**) compares all the annotated SMases enzymes from *E. histolytica* and *T. vaginalis* in UniProt or GenBank. The light blue box indicates the *T. vaginalis* enzymes, and the light pink box indicates the aSMase6 from *E. histolytica*. The evolutionary history was inferred using the maximum likelihood method and JTT matrix-based model [51]. The tree with the highest log likelihood (−19,141.69) is shown. The percentage of trees on which the associated taxa clustered is shown next to the branches. The heuristic search’s initial tree was obtained automatically by applying the Neighbor-Join and BioNJ algorithms to a matrix of estimated pairwise distances using the JTT model. Then, the topology with a superior log likelihood value was selected. The tree is drawn to scale, with branch lengths measured in the number of substitutions per site. This analysis involved 22 amino acid sequences. There was a total of 487 positions in the final dataset. Evolutionary analyses were conducted in MEGA11 [31]. Panel (**B**) Evolutionary analysis considering all annotated enzymes in UniProt from Bacillus cereus (purple box), *Giardia intestinalis* (orange box), *Acanthamoeba castellanii* (yellow box), *Dictyostelium discoideum* (dark green box), *E. hystolitica* (pink box), and *T. vaginalis* (light blue box). The dashed black arrow indicates the position of aSMase6 of *E. histolytica*. As in Panel A, the evolutionary history was inferred using the maximum likelihood method and JTT matrix-based model [51]. The tree with the highest log likelihood (−47,796.15) is shown. The percentage of trees on which the associated taxa clustered is shown next to the branches. The heuristic search’s initial tree was obtained automatically by applying the Neighbor-Join and BioNJ algorithms to a matrix of estimated pairwise distances using the JTT model. Then, the topology with a superior log likelihood value was selected. The tree is drawn to scale, with branch lengths measured in the number of substitutions per site. This analysis involved 60 amino acid sequences. There was a total of 487 positions in the final dataset.

**Figure 7 pathogens-14-00032-f007:**
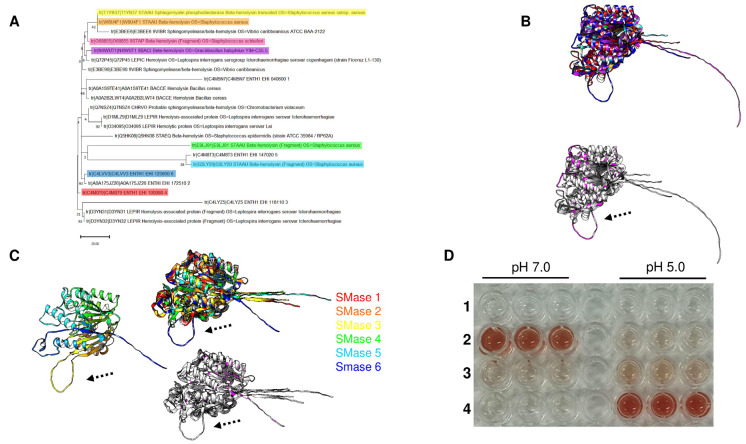
The aSMase6 enzyme exhibits features of hemolysins and experimentally lyses human red cells. Panel (**A**) is a phylogenetic reconstruction using examples of bacterial hemolysins in comparison with SMase 6, which exhibits hemolytic activity *in vitro*, and SMase 4, which does not exhibit hemolytic activity *in vitro* (see Panel (**D**)). The phylogenetic analysis shows closeness to *S. aureus* hemolysins. The evolutionary history was inferred using the maximum likelihood method and JTT matrix-based model [51]. The tree with the highest log likelihood (−25,879.64) is shown. The percentage of trees in which the associated taxa clustered together is shown below the branches. The initial tree for the heuristic search was obtained automatically by applying the Neighbor-Join and BioNJ algorithms to a matrix of estimated pairwise distances using the JTT model, and then the topology with a superior log likelihood value was selected. The tree is drawn to scale, with branch lengths measured in the number of substitutions per site. This analysis involved 23 amino acid sequences. There were a total of 1221 positions in the final dataset. The percentage of trees on which the associated taxa clustered together is shown below the branches. Initial tree(s) for the heuristic search were obtained by applying the Neighbor-Joining method to a matrix of pairwise distances estimated using the JTT model. The tree is drawn to scale, with branch lengths measured in the number of substitutions per site. This analysis involved 18 amino acid sequences. There were a total of 1254 positions in the final dataset. In Panel (**B**), a structural comparison of the AlphaFold2 models of SMase4 (in red, accession number AF-C4M0T9-F1) and SMase6 (in blue, accession number AF-C4LV33-F1) in comparison with Staphylococcus shleiferi (UniProt 068855, in pink [AlphaFold accession number: AF-O68855-F1])), S. aureus (UniProt E9LJ91 [in green, AF-E9LJ91-F1], G3LY20 [in cyan, AF-G3LY20-F1], T1YB37 [in yellow, AF-T1YB37-F1] and W8U4F1 [in orange, AF-W8U4F1-F1]), and *Gracilibacillus halophilus* (UniProt N4WUT, in purple, AF-N4WUT-F1) hemolysins (RMSD 3.03 Å and TM score 0.74416 normalized by the length of the shorter structure). In Panel (**C**), the comparison between all *E. histolytica* aSMase enzymes using AlphaFold2 models retrieved from UniProt is shown. The color code for each one is indicated in the figure. The first model, from left to right, is the aSMase6 in the rainbow color scheme. The next model is the overlay of the six enzymes (RMSD of 2.41Å and a TM score of 0.86749 normalized by the length of the shorter structure). In the next model, the conserved regions are indicated in magenta. The final model shows the most different features conserved compared to the bacterial hemolysins. This loop contains the sequence QLRPKKDKLKPKTTIKQPSDGFN. AlphaFold2 accession numbers: aSMase1: AF-C4M5N7-F1; aSMase2: AF-A0A175JZ26-F1; aSMase3: AF-C4LYZ5-F1; and aSMase5: AF-C4M8T3-F1. In the structural alignments, when sequences were compared, the conserved sequences are shown in magenta, determined with the UCSF Chimera Clustal Omega plug-in. In Panel (**D**), recombinant SMase UniProt C4M0T9 (SMase4) and C4LVV3 (SMase6) were expressed in *E. coli* and purified and tested for hemolytic activity. Reactions were carried out in triplicates. In the figure, the upper lines indicate the wells where the reactions were carried out at pH 7.0 or pH 5.0. Control reactions with erythrocytes in the wells are indicated with 1, but no enzyme was added. In wells indicated with 2, erythrocytes were exposed to commercially available *B. cereus* nSMase. In wells indicated with 3, recombinant aSMase4 was used. In the well indicated with 4, recombinant aSMase6 was used.

**Figure 8 pathogens-14-00032-f008:**
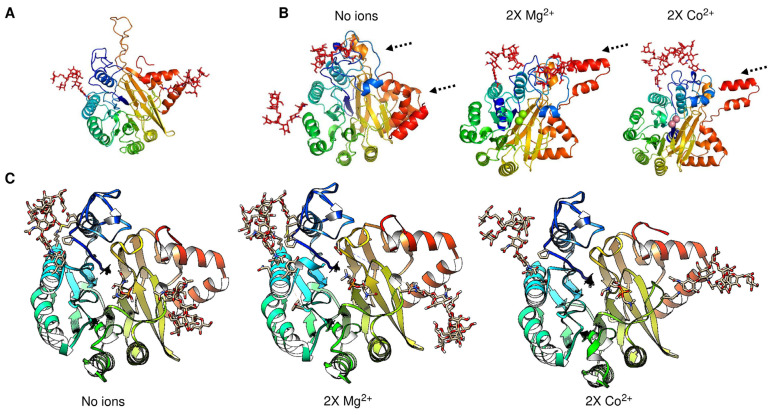
aSMase4 and aSMase6 differ in an internal loop containing several charged residues. In Panel (**A**), the glycosylated aSMase6 model is shown compared to the following panels. Panel (**B**) models of glycosylated aSMase4 with no ions or containing two Mg^2+^ ions or two Co^2+^ ions. Dashed arrows indicate the position of the N- and C-terminal ends, showing structural variation depending on the ion bound. Panel (**C**) structure and docking result of the aSMase6 protein, glycosylated but lacking the charged loop. The absence or presence of ion modeled in the structure is indicated. In all instances, the position of the substrate is nearly identical.

**Figure 9 pathogens-14-00032-f009:**
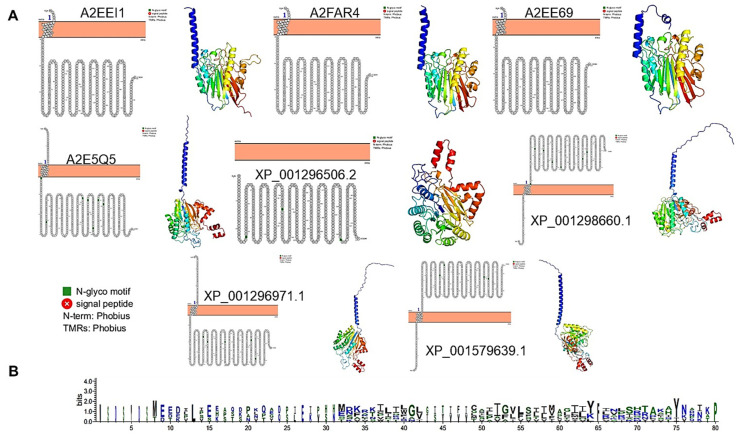
*T. vaginalis* SMases show a different topology than the *E. histolytica* enzymes. Panel (**A**) Protter topology and post-translational modification predictions and AlphaFold3 models of all annotated *T. vaginalis* enzymes. Protter legend is indicated in the figure. Each enzyme accession number (either UniProt or GenBank) is indicated. The cell membrane is shown as a light orange bar. The predicted signal peptide is shown in red. Predicted glycosylation sites are displayed in green. In the structural model, a rainbow color scheme is used. The N-terminal domains showing a similar membrane-associated domain were analyzed for sequence homology. As shown in Panel (**B**), no conserved residues or sequences were found, except for conserved tyrosine, suggesting that these enzymes may not contain a signal peptide.

**Figure 10 pathogens-14-00032-f010:**
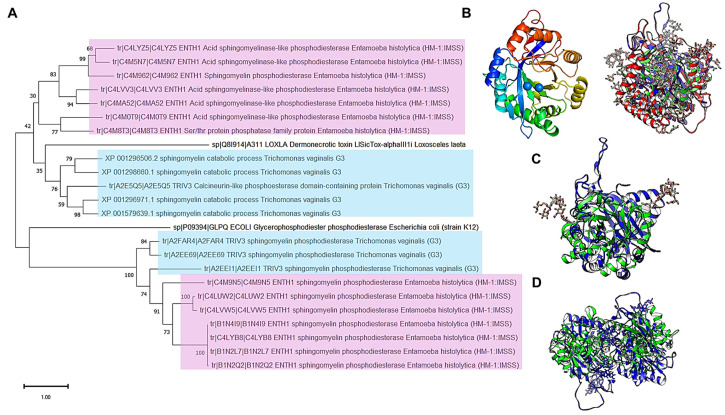
In the search for the ancestry of SMase enzymes, a conserved core element may be the common element in all SMases. Panel (**A**) Evolutionary history of SMases from *E. histolytica* (in pink), *T. vaginalis* (in light blue), and two enzymes, the putative ancient glyercerophosphodiester phosphodiesterase from *E. coli* and an example of dermonecrotic toxin (SMase D) from the spider *Loxosceles laeta*, was inferred using the maximum likelihood method and the JTT matrix-based model [51]. The tree with the highest log likelihood (−19,199.42) is shown. The percentage of trees on which the associated taxa clustered together is shown below the branches. The heuristic search’s initial tree was obtained automatically by applying the Neighbor-Join and BioNJ algorithms to a matrix of estimated pairwise distances using the JTT model. Then, the topology with a superior log likelihood value was selected. The tree is drawn to scale, with branch lengths measured in the number of substitutions per site. This analysis involved 24 amino acid sequences. In Panel (**B**), the structural features of the available structures or AlphaFold3 models generated in this work were compared with the structure of the two putative minimal SMase enzymes. On the left is the rainbow color scheme of the model from *L. laeta* SMase D, and on the right is structural alignment with human aSMase (PDB 5I81, in red), with *E. histolytica* glycosylated and with two Mg^2+^ ions nSmase6 (in blue) and in green, the *L. laeta* SMase D model (RMSD of 4.49 Å, TM score normalized by the length of the shorter structure of 0.58819) is shown. In Panel (**C**), the structural alignment of the *L. laeta* SMase D (in green) and aSMase6 (glycosylated and with two Mg^+2^ ions) of the *E. histolytica* enzyme (in blue) shows an RMSD of 5.37 and a TM score normalized by the length of the shorter sequence of 0.46147. Panel (**D**) comparison between the crystal structure of *E. coli* glyercerophosphodiester phosphodiesterase (PDB 1YDY, in green) with the predicted dimer of *E. histolytica* aSMase6 (glycosylated and with two Mg^+2^ ions), showing an RMSD of 7.84 and a TM score normalized by the length of the shorter structure of 0.25975.

## Data Availability

The PDB files used in this work are provided as a zip file in the same order as Table 1 and then as shown for *T. vaginalis* in Appendix A. Appendix A are provided. Additional data will be freely available upon reasonable request.

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
