# Peer review of "Theoretical Study of Sphingomyelinases from Entamoeba histolytica and Trichomonas vaginalis Sheds Light on the Evolution of Enzymes Needed for Survival and Colonization"

_pathogens, 2025, doi:10.3390/pathogens14010032_

Round 1

Reviewer 1 Report

Comments and Suggestions for Authors

The manuscript exemplifies a sophisticated integration of computational biology, structural biochemistry, and evolutionary analysis, providing a multidimensional perspective on the intricate biology of acid sphingomyelinases in parasitic systems. It makes a substantial contribution to our understanding of sphingomyelinase (SMase) evolution by uncovering a conserved structural core alongside functional diversification. Particularly noteworthy are the novel hypotheses regarding SMase secretion mechanisms in Trichomonas vaginalis and their potential roles in extracellular vesicle dynamics—a promising frontier with profound implications for elucidating pathogen-host interactions.

Regarding the minor comment on Table 1's Vina scores:

1.The authors should explicitly clarify the following aspects of the binding affinity analysis: Confirm and explicitly state the unit for the Vina scores (typically kcal/mol).

2.Provide a detailed interpretation of the relatively low binding affinities

Author Response

Response letter to reviewers

We appreciate the positive feedback and comments on our research and the opportunity to provide a revised manuscript according to the observations, corrections, and suggestions for our work. All the modifications made to the manuscript are highlighted in yellow in the provided manuscript.  The responses are indicated in red.

Once again, thank you so much for the opportunity to move forward with our research.

The manuscript exemplifies a sophisticated integration of computational biology, structural biochemistry, and evolutionary analysis, providing a multidimensional perspective on the intricate biology of acid sphingomyelinases in parasitic systems. It makes a substantial contribution to our understanding of sphingomyelinase (SMase) evolution by uncovering a conserved structural core alongside functional diversification. Particularly noteworthy are the novel hypotheses regarding SMase secretion mechanisms in Trichomonas vaginalis and their potential roles in extracellular vesicle dynamics—a promising frontier with profound implications for elucidating pathogen-host interactions.

Dear Reviewer, thank you so much for the positive insight. Indeed, due to the lack of tools for generating enough protein directly from the parasite to assess the true post-translational modifications, we exploited the capabilities of AlphaFold3 regarding generating models with such modifications and more. Using the data presented here, a plethora of new experimental approaches will be undertaken in our group and hopefully in other research groups.

Regarding the minor comment on Table 1's Vina scores:

  1. The authors should explicitly clarify the following aspects of the binding affinity analysis: Confirm and explicitly state the unit for the Vina scores (typically kcal/mol).

Thank you for the observation; we confirm that the units are kcal/mol Lines477-478), and in Table 1, we have included the units for the Vina scores. Also, we corrected to “Vina score” in the same table.

  1. Provide a detailed interpretation of the relatively low binding affinities

Thank you so much for the observation. We did not include this issue that any docking simulation has with molecules such as sphingomyelin; as observed, it is highly relevant. The major issue with this substrate is the presence of 20 rotamer sections in the molecules, which strongly reduces the Vina Score; additionally, the cavity in this enzyme is small, and the Vina scores are low. However, as stated in the manuscript, no other relevant cavities were detected. The data is consistent with the metal ion residues and the position of the substrate inside the structure.  In lines 513-519, we included the following: “The limitation of using CB-Dock2 lies in the Vina scoring function, a weighted sum of atomic interactions. The models used in this work include charged glycan moieties, and their steric, hydrophobic, and hydrogen bonding interactions are calculated. To account for entropic penalties, the Vina score is reduced when the interactions are adjusted based on the number of rotatable bonds in the sphingomyelin substrate [50]. The data, however, is consistent with the metal-bound cavities, and no other cavities were detected in the protein, providing confident information to target key residues for mapping the active site in these enzymes.”

Corrections indicated by the other reviewers are also included in the manuscript.

Again, thank you so much for the insight.

Reviewer 2 Report

Comments and Suggestions for Authors

In this manuscript prepared by Ramírez-Montiel and colleagues, utilizing Alphafold and other bioinformatic tools, the authors explored and analyzed the structural and functional characteristics of putative sphingomyelinases from two pathogenic organisms.  The enzymatic activity of SMases from E. histolytica are also tested experimentally in vitro.  Overall, the manuscript is well-structured and presents a scientifically convincing story.

Comments that may further strengthen this manuscript are provided as follows:

Minor issues:

  1. Could the authors also show the confidence metrics for all the Alphafold3 models for E. histolytica SMase6 as for T. vaginalis models. Since Alphafold3 server generates 5 models for each prediction, could the authors specify which model is used and justify the reason? I would also recommend the authors to make a table to list all prediction details and probably submit all original prediction .cif files (both used and unused).
  2. When predicting the models with different ions (especially with two ions), what is the confidence value for each ion and metal binding sites?
  3. When predicting the join models of human cathepsin D/cysteine protease candidates and aSMase6 from E. histolytica, what stoichiometry did the authors use? Monomer to monomer, or dimer to dimer? Different prediction stoichiometry may give different models. Please specify and justify.
  4. Would T. vaginalis SMases also form homodimer? If they do, could the authors also explore the dimer interaction modes of these topologically different SMases?
  5. Wrong citations: Line 295, Line 299 (switched). Please also carefully check all other citations.
  6. Grammatical or typographical errors:

       1. Line 22, 23, 26,28: please use the verb tense consistently for “show”, “show”, “showed” and “revealed”.

        2. Line 486: asses —> assesses.

Suggestions:

  1. It will be more informative if the activity results can be shown with mutants lacking the loop which the authors proposed to be involved in hemolytic activity.
  2. As now the source code is released (https://github.com/google-deepmind/alphafold3), the authors could use Alphafold3 to predict substrates binding modes and more predictions without limitations.
  3. It is worth mentioning that the use of AlphaFold in combination with Foldseek may assist in mining protein candidates with low sequence homology but having high structural similarity. (PMID: 37386183 PMID: 38948817, PMID: 39187718)

4. Another highly accurate protein structure prediction tool, RoseTTAFold-All-Atom, developed by David Baker’s laboratory, is also worth trying  (https://github.com/baker-laboratory/RoseTTAFold-All-Atom)

Author Response

Response letter to reviewers

We appreciate the positive feedback and comments on our research and the opportunity to provide a revised manuscript according to the observations, corrections, and suggestions for our work. All the modifications made to the manuscript are highlighted in yellow in the provided manuscript.  The responses are indicated in red.

Once again, thank you so much for the opportunity to move forward with our research.

In this manuscript prepared by Ramírez-Montiel and colleagues, utilizing Alphafold and other bioinformatic tools, the authors explored and analyzed the structural and functional characteristics of putative sphingomyelinases from two pathogenic organisms.  The enzymatic activity of SMases from E. histolytica are also tested experimentally in vitro.  Overall, the manuscript is well-structured and presents a scientifically convincing story.

Thank you so much for the positive insight and the opportunity to improve our research. We address your observations and corrections in the following lines.

Comments that may further strengthen this manuscript are provided as follows:

Minor issues:

Could the authors also show the confidence metrics for all the Alphafold3 models for E. histolytica SMase6 as for T. vaginalis models. Since Alphafold3 server generates 5 models for each prediction, could the authors specify which model is used and justify the reason? I would also recommend the authors to make a table to list all prediction details and probably submit all original prediction .cif files (both used and unused).

Thank you so much for the observation. Indeed, we missed that information from the Methods section. We employed the rank 1 model in all instances since it contains the highest predicted template modeling (pTM) score. A pTM score above 0.5 means the overall predicted fold for the complex might be similar to the true structure. In all instances, the model without post-translational modifications rendered scores above 0.8; even with such modifications, the overall pTM score was not severely affected. The models approximate the true structure; the protein modeling will never replace the experimental determination of the protein structure. However, it is a tool for assessing the observed experimental data and further designing experimental approaches to test the generated hypothesis. In this case, we aim to target the production and establish a collaboration with a colleague focusing on protein crystallography.

Additionally, we have included this information in lines 153-157, and the quality of the models (pTM score) of all models used are provided in Supplementary Table 1 in the same order as in Table 1. In the supplementary files, we included the five .cif files for the provided models, and a legend was added to line 985.

When predicting the models with different ions (especially with two ions), what is the confidence value for each ion and metal binding sites?

Thank you for the question. In all instances, the metal ion prediction ranges from a plDDT>90 or 90>plDDT>70; no major deviation from this range was observed. We reply to this question regarding plDDT since this is a parameter for local model variation in quality. Structural alignments confirmed this regardless of the post-translational modifications, and it was consistent with the human crystallographic structure.

When predicting the join models of human cathepsin D/cysteine protease candidates and aSMase6 from E. histolytica, what stoichiometry did the authors use? Monomer to monomer, or dimer to dimer? Different prediction stoichiometry may give different models. Please specify and justify.

Thank you so much for the question and observation. We employ monomeric forms of both proteins to show in the manuscript since the dimeric form of the proteins, as shown in Figure 4, the region encompassing the C-terminal end, is buried inside the interphase of the protein as for the human crystallographic structure. We did a model using the dimeric form of both proteins, but in that instance, the interaction is limited to a couple of residues on the opposite face of the protein. Thus, in the final version of the manuscript, we only included the monomeric form of each protein. This has been stated in the manuscript in lines 381-383.

Would T. vaginalis SMases also form homodimer? If they do, could the authors also explore the dimer interaction modes of these topologically different SMases?

Thank you for your observation. Perhaps these proteins have a different conformation, they may act in vivo as monomers based on the model for one of the proteins (UniProt ASEEI1):

(Please see Figure in attached file)

For the model shown, the quality parameters shift to a pTM of 0.53, and as shown, several regions, especially the metal ions bound (spheres), to the proteins. The rest of the proteins showing a different topological feature, such as the one shown here, have the same dimeric forms. That is why we did not include this information in the final version of the manuscript, more biochemical work is needed to assess the topology of these enzymes. 

Wrong citations: Line 295, Line 299 (switched). Please also carefully check all other citations.

Thank you for pointing this out. We have corrected the reference list, the following reference was located in the wrong place:  Qiu H, Edmunds T, Baker-Malcolm J, Karey KP, Estes S, Schwarz C, Hughes H, Van Patten SM. Activation of human acid sphingomyelinase through modification or deletion of C-terminal cysteine. J Biol Chem. 2003 Aug 29;278(35):32744-52. doi: 10.1074/jbc.M303022200. (reference 44), now is corrected in lines 1124-1126.

Grammatical or typographical errors:

  1. Line 22, 23, 26,28: please use the verb tense consistently for “show”, “show”, “showed” and “revealed”.

Thank you so much for the observation. We have corrected the abstract accordingly.

  1. Line 486: asses —> assesses.

Thank you so much for the observation; this has been corrected.

Suggestions:

It will be more informative if the activity results can be shown with mutants lacking the loop which the authors proposed to be involved in hemolytic activity.

We fully agree with the reviewer regarding these important experiments. However, since the journal provided five days to respond, we are unable to comply with the time frame. We kindly request the reviewer to allow us to perform the experiments both in vitro and in vivo for a future follow-up paper. We consider that we can perform both types of experiments to assess the effect of such mutation on the virulence of E. histolytica.

Now that the source code is released (https://github.com/google-deepmind/alphafold3), the authors could use Alphafold3 to predict substrate binding modes and make other predictions without limitations.

Thank you very much for the suggestion. New tools are being released that will allow different predictions to be made, such as AlphaFold3, which is now open source. AlphaFold3 was released two weeks ago in GitHub, but it takes time to become familiar with the program to take advantage of it and obtain new results in another manuscript. The tool we used to generate data for this work is pertinent, so we kindly request that you keep the data we have in this paper, and we will continue to use this useful tool now that it runs locally.

It is worth mentioning that the use of AlphaFold in combination with Foldseek may assist in mining protein candidates with low sequence homology but having high structural similarity. (PMID: 37386183 PMID: 38948817, PMID: 39187718)

Thank you for the suggestion. We have included it in lines 833-836 and added the reference for FoldSeek (lines 1147-1148).

  1. Another highly accurate protein structure prediction tool, RoseTTAFold-All-Atom, developed by David Baker’s laboratory, is also worth trying (https://github.com/baker-laboratory/RoseTTAFold-All-Atom)

Thank you for the suggestion. We have used the tool, but since we wanted to explore the structural features using post-translational modifications, which turn out to be faster and comparably more accurate in AlphaFold3. We kindly request that the data presented here be maintained using the AlphaFold3 tool.

Reviewer 3 Report

Comments and Suggestions for Authors

Fátima Berenice Ramírez-Montiel et al. proceeded with the complementary research about the sphingomyelinases from Entamoeba histolytica and Trichomonas vaginalis by bioinformatical tools, such as alphafold3. They completed the manuscript entitled ‘Theoretical study of sphingomyelinases from Entamoeba histolytica and Trichomonas vaginalis sheds light on the evolution of enzymes needed for survival and colonization.’ The manuscript demonstrated a series of bioinformatical studies on aSMases, including multiple sequence alignments and structural predictions. However, the analysis of the structure comparison is unreliable, because almost all results are carried out in silicon, in particular the structural predictions,  even the predicted structures are very similar to the crystal structure of human aSMase, I strongly suggest at least one experimental structure (crystallography or CD) should be included in the manuscript to make the conclusion more reliable.

I have some research questions that I would like to know.

I'm curious about the activity of the enzyme that was stimulated by Mg2+ but inhibited by Co2+. I found that in the crystal structure of human aSMase, Zn2+ is buried by H; is this the zinc-finger motif?  Mg2+, in my opinion, cannot substitute Zn2+ in a zinc-finger motif. While for the different Mg2+ and Co2+, I cannot image the huge impact on the enzyme structure, and the two ions bind to similar positions in the predicted structures. If Mg2+ and Co2+ can form different coordination fields with residues to work, then they should act on different residues.

Furthermore, if Co2+ inhibits activity, what happens to Mn2+ and Ni2+?

Did the author run the structural search on the Dali server?  The same structure should perform the same function. As a result, I believe this would be useful in studying the evolution of enzymes in nature.

Some details in the manuscript could also be improved. For example, for the structure depicted in the figure, the author should show the exact residues involved and use different colors to represent distinct atoms (Figures 5, 6, and 7). The presentation of the article should be accurate and concise, and avoid repetitive descriptions.

Author Response

Response letter to reviewers

We appreciate the positive feedback and comments on our research and the opportunity to provide a revised manuscript according to the observations, corrections, and suggestions for our work. All the modifications made to the manuscript are highlighted in yellow in the provided manuscript.  The responses are indicated in red.

Once again, thank you so much for the opportunity to move forward with our research.

Fátima Berenice Ramírez-Montiel et al. proceeded with the complementary research about the sphingomyelinases from Entamoeba histolytica and Trichomonas vaginalis by bioinformatical tools, such as alphafold3. They completed the manuscript entitled ‘Theoretical study of sphingomyelinases from Entamoeba histolytica and Trichomonas vaginalis sheds light on the evolution of enzymes needed for survival and colonization.’ The manuscript demonstrated a series of bioinformatical studies on aSMases, including multiple sequence alignments and structural predictions. However, the analysis of the structure comparison is unreliable, because almost all results are carried out in silicon, in particular the structural predictions,  even the predicted structures are very similar to the crystal structure of human aSMase, I strongly suggest at least one experimental structure (crystallography or CD) should be included in the manuscript to make the conclusion more reliable.

Thank you for the insight. We agree that experimental structural validation is mandatory to truly assess the mechanistic dynamics in enzymes. However, we are not a crystallography laboratory; thus, performing such experiments at the moment is out of our reach and requires establishing collaboration for such a task. However, we believe that protein modeling is a powerful tool for generating evolutionary insight and designing biochemical experiments for the future, and the generated data for this work is pertinent. The models presented here are of good quality, and according to the recommendation of another reviewer, we have added a supplementary table including the model quality data.

I have some research questions that I would like to know.

I'm curious about the activity of the enzyme that was stimulated by Mg2+ but inhibited by Co2+. I found that in the crystal structure of human aSMase, Zn2+ is buried by H; is this the zinc-finger motif?  Mg2+, in my opinion, cannot substitute Zn2+ in a zinc-finger motif. While for the different Mg2+ and Co2+, I cannot image the huge impact on the enzyme structure, and the two ions bind to similar positions in the predicted structures. If Mg2+ and Co2+ can form different coordination fields with residues to work, then they should act on different residues.

Thank you so much for your question. Regarding the binding of Zn ions in the human enzyme is not a zinc finger; the ions have a trigonal bipyramidal geometry and are coordinated by Asp278 (bridges the two ions), His457, and His459 (are at the axial position) in the human enzyme numbering (Zhou et al., 2016, doi: 10.1038/ncomms13082). Now, the coordination in the E. histolytica enzyme involves the same residues. The mechanism is similar, as reported previously by Ramírez-Montiel et al., 2019 (doi: 10.1371/journal.ppat.1008016), that showed that the secreted SMase activity by the parasite shows Mg activation, no effect by Ca, Zn, Mn and EGTA showed a 25%, 30% to 50% respectively decrease in activity while Co inhibited the enzyme at very low concentrations (doi: 10.1371/journal.ppat.1008016). We must stress out that this is the activity analyzed directly from the parasite. For a follow-up paper, we are designing an alanine scan to mutate several surrounding residues at the coordination sites to define, based on the models presented here, the residues needed for both activity, metal coordination and enzyme processing.

Furthermore, if Co2+ inhibits activity, what happens to Mn2+ and Ni2+?

Thank you so much for your question. As stated in the previous response, the Mn effect reported by Ramírez-Montiel and colleagues is a reduction in activity, Ni has not been tested.

Did the author run the structural search on the Dali server?  The same structure should perform the same function. As a result, I believe this would be useful in studying the evolution of enzymes in nature.

Thank you for your suggestion. We searched for the aSMase6 enzyme from E. histolytica in the Deli server, and the relevant hits were derived from the murine enzyme; the rest of the hits were hydrolases such as beta-N-acetylglucosaminidase and other glycosidases. So, we did not follow this strategy because the diversity of glycosidases found did not help us further study the evolutionary path of these enzymes. We had better hits with FoldSeek, but many of the hits need biochemical characterization to assess the true activity of these enzymes, which we will characterize soon.

Some details in the manuscript could also be improved. For example, for the structure depicted in the figure, the author should show the exact residues involved and use different colors to represent distinct atoms (Figures 5, 6, and 7). The article should also be presented accurately and concisely and avoid repetitive descriptions.

Thank you for the comment. However, regarding the figures, the important residues are indicated in all instances. We have revised the manuscript for the article presentation, and we could not identify inaccurate or lengthy sections. Perhaps the reviewer could help us identify the sections that need clarification. Thank you for your comments on how to improve this manuscript.

Round 2

Reviewer 1 Report

Comments and Suggestions for Authors

Well done, nice job.

Author Response

Thank you so much for your positive insight. 

We hope the manuscript will pave the way for new understanding of these enzymes. 

Best regards and best wishes for this 2025. 

Reviewer 2 Report

Comments and Suggestions for Authors

All comments are properly resolved in the revised manuscript. 

Author Response

Thank you so much for the positive insight. We hope this paper will pave the way for future experiments to advance our understanding or SMases. 

Best regards, and all the best for 2025 for you and your loved ones. 

Reviewer 3 Report

Comments and Suggestions for Authors

Thank you for your responses to my questions. I appreciate the effort in preparing the updated version of the manuscript. However, there are a few unclear descriptions and minor issues that should be addressed:

  1. Figure 4, Panel B: The graph does not include the Mg²⁺ and Co²⁺ ions, even though they are mentioned in the figure legend. These ions should be clearly highlighted in the graph to ensure consistency with the legend.
  2. Line 302: The manuscript mentions that a water molecule is required for human sSMase activation. Could this circumstance be accounted for in AlphaFold 3? 
  3. Figure 6, Panel A: The disulfide bond is typically depicted in yellow, while residues shown in stick representation are colored by atoms. For example, in Panel B, the ‘O’ atom of Glu 69 interacts with Cys, the ‘O’ is generally colored in red. These color conventions should be clearly defined and consistently applied.
  4. Lines 188 and 193: There are numbering errors in these lines that should be corrected.
  5. Figure 13: The graph's size is inappropriate and should be adjusted to better fit the layout of the manuscript.
  6. General comments on manuscript length and conciseness:
    • The manuscript is still overly long, redundant, and lacks conciseness. The Introduction section could be streamlined to provide a more focused background and clearly set the stage for the research. For instance, line 99 appears unnecessary and could be removed.
    • In the Results section, Figures 1, 2, and 3 could be merged into a single figure, with some panels moved to the supplementary material. This approach could be applied to other figures as well, reducing the overall length and improving readability.

I hope these suggestions help make the manuscript clearer, more concise, and easier to read.

Author Response

Thank you for your responses to my questions. I appreciate the effort in preparing the updated version of the manuscript. However, there are a few unclear descriptions and minor issues that should be addressed:

Thank you so much for your observations. In the revised manuscript we indicate the changes with cyan highlight.

Figure 4, Panel B: The graph does not include the Mg²⁺ and Co²⁺ ions, even though they are mentioned in the figure legend. These ions should be clearly highlighted in the graph to ensure consistency with the legend.

Thank you so much for the observation. We have included a ribbon version of the same structural alignment slightly tilted to show the small deviation that is observed in the position of both metal ions using the front view of the protein (middle panel now in Figure 4 panel B). Also, the figure legend was complemented regarding this new figure.

Line 302: The manuscript mentions that a water molecule is required for human sSMase activation. Could this circumstance be accounted for in AlphaFold 3?

Thank you for the comment. No, AlphaFold3, even in the open-source version, cannot model yet the water molecules inside the structure, only ligands, post-translational modifications and metal ions.

Figure 6, Panel A: The disulfide bond is typically depicted in yellow, while residues shown in stick representation are colored by atoms. For example, in Panel B, the ‘O’ atom of Glu 69 interacts with Cys, the ‘O’ is generally colored in red. These color conventions should be clearly defined and consistently applied.

Thank you for your observation. However, we used the conventional coloring as suggested, but the color is not visible in the version of the figure. We kindly request to keep the coloring as presented to avoid unclear image. As an example, here we show the interaction of aSMase6 (left) and human cathepsin D (right) using the yellow coloring for the C-terminal end cysteines. This is the reason behind using the red and green coloring. Also, the heteroatom coloring is barely visible.

(Please check the attached file for the image)

Lines 188 and 193: There are numbering errors in these lines that should be corrected.

Thank you for the correction, we have addressed this in the new version of the manuscript.

Figure 13: The graph's size is inappropriate and should be adjusted to better fit the layout of the manuscript.

Thank you for the comment, we have enlarged the figure to enhance the readability of the legends on the phylogenetic analysis. Thank you for pointing out that this was needed.

General comments on manuscript length and conciseness:

The manuscript is still overly long, redundant, and lacks conciseness. The Introduction section could be streamlined to provide a more focused background and clearly set the stage for the research. For instance, line 99 appears unnecessary and could be removed.

Thank you for pointing this out. We have removed the cited line and the introduction was revised. The changes are indicated in cyan, the parts that will be removed are crossed out besides line 99 that has been removed as indicated by the reviewer.

In the Results section, Figures 1, 2, and 3 could be merged into a single figure, with some panels moved to the supplementary material. This approach could be applied to other figures as well, reducing the overall length and improving readability.

Thank you for the suggestions. In the case of the figures indicated by the reviewer, Figures 1, 2, and 3 were sent to supplementary figures. As the reviewer points out, the strong suite of the paper is the structural analysis, and the three figures reduce the impact of the rest of the content of the manuscript. All legends have been corrected. However, the rest of the manuscript, we did not move more figures to the supplementary figures, since each one addresses a different aspect of the SMases under study, we begin by validating the models against the human enzyme, then we move to characterizing the relevant C-terminal end processing that is a key modification of the aSMase to be secreted and is essential for the membrane repair mechanism previously described, then we address the binding of the substrate, then we address the evolutionary relationship with hemolytic activity and finally we show that in another protist the same enzymes, even though they have the same folding for the active site, show a different topology. Also, we suggest that the basic folding of the active site is conserved by the earliest ancestors. So, we think that the suggested figures needed to be in supplementary material but not the rest. We kindly request you to keep the rest of the figures as presented originally in this manuscript to present in a continuous flow the content of this manuscript. Thank you for your insight.

I hope these suggestions help make the manuscript clearer, more concise, and easier to read.

We thank you again for the suggestions and positive insight into this manuscript.

Round 3

Reviewer 3 Report

Comments and Suggestions for Authors

Thank you for addressing my questions. However, I still have a concern regarding AlphaFold 3. You mentioned that water molecules cannot be accounted for in the structural predictions generated by AlphaFold 3. This raises an issue because water molecules often play critical roles in enzymatic activation or recognition, particularly around metal ions, where they frequently act as bridges connecting the metal ion to surrounding residues.

Moreover, you pointed out that a water molecule is essential for the activation of human aSMase (in line 272-276), an enzyme homologous to the two enzymes you studied. This makes me worry that your predicted models generated through AlphaFold 3 may have missed the functional effects of water molecules. Would it be acceptable to discuss the role of the metal ion separately in your analysis?

Author Response

Thank you for addressing my questions. However, I still have a concern regarding AlphaFold 3. You mentioned that water molecules cannot be accounted for in the structural predictions generated by AlphaFold 3. This raises an issue because water molecules often play critical roles in enzymatic activation or recognition, particularly around metal ions, where they frequently act as bridges connecting the metal ion to surrounding residues.

Moreover, you pointed out that a water molecule is essential for the activation of human aSMase (in line 272-276), an enzyme homologous to the two enzymes you studied. This makes me worry that your predicted models generated through AlphaFold 3 may have missed the functional effects of water molecules. Would it be acceptable to discuss the role of the metal ion separately in your analysis?

Thank you for your observation. None of the available protein modelers cannot predict water molecules inside a protein structure. As you point out, there is a strong limitation on protein modeling to predict the position of water bridges, salt bridges, and the presence of other salt ions inside a protein structure accurately. However, as pointed out in lines 318-320 of the manuscript, we highlight the importance of Figure 1, as you suggested, that the crystal structure (PDB 5I81) of the human acid sphingomyelinase, AlphaFold3 learning could predict the position of the metal ions in the same allocation as the zinc ions in the human enzyme, which is a major breakthrough for protein modeling. AlphaFold software has been trained with the currently available protein structures from PDB, and by the time AlphaFold3 was released, over 190,000 protein structures were used for the training process of this software. So, the prediction used based on the scope of the paper shows unambiguously that the primitive fold of the sphingomyelinases is conserved, and the metal ion pocket is conserved across these enzymes, which is a major advance in the understanding of these enzymes and paves the way for future experiments as stated in the previous revisions of this paper. Also, as commented before regarding the need to experimentally obtain the structure of aSMase6, which is a big challenge, since the growth of the parasite to obtain the native form of the protein is no easy task, since the growth conditions limit the volume of cells that can be grown and the amount of protein secreted. Up to date, we haven’t been able to quantify the exact amount of enzyme that is secreted. Nevertheless, we are designing a method to obtain as much protein as possible which may help us to obtain enough protein for crystallography attempts.

Now, with the previous lines and the biochemical data provided in our previous research, we kindly request to maintain the manuscript as it is and we have added the following lines: With the data presented in Figure 1, AlphaFold3 training is able to predict the position of Mg2+ in aSMase6 in the same pocket as in human aSMase, which allows to further explore the biochemical data gathered for this enzyme with a structural point of view.

This line is highlighted in green and is in lines 320-323.

We hope this is acceptable for the reviewer.

Best regards.